# ICE1 promotes the link between splicing and nonsense-mediated mRNA decay

Thomas D Baird[1], Ken Chih-Chien Cheng[2], Yu-Chi Chen[2], Eugen Buehler[2], Scott E Martin[2], James Inglese[2], J Robert Hogg[1]*

[1]Biochemistry and Biophysics Center, National Heart, Lung, and Blood Institute, National Institutes of Health, Bethesda, United States; [2]National Center for Advancing Translational Sciences, National Institutes of Health, Rockville, United States

**Abstract** The nonsense-mediated mRNA decay (NMD) pathway detects aberrant transcripts containing premature termination codons (PTCs) and regulates expression of 5–10% of non-aberrant human mRNAs. To date, most proteins involved in NMD have been identified by genetic screens in model organisms; however, the increased complexity of gene expression regulation in human cells suggests that additional proteins may participate in the human NMD pathway. To identify proteins required for NMD, we performed a genome-wide RNAi screen against >21,000 genes. Canonical members of the NMD pathway were highly enriched as top hits in the siRNA screen, along with numerous candidate NMD factors, including the conserved ICE1/KIAA0947 protein. RNAseq studies reveal that depletion of ICE1 globally enhances accumulation and stability of NMD-target mRNAs. Further, our data suggest that ICE1 uses a putative MIF4G domain to interact with exon junction complex (EJC) proteins and promotes the association of the NMD protein UPF3B with the EJC.

DOI: https://doi.org/10.7554/eLife.33178.001

*For correspondence:
j.hogg@nih.gov

Competing interests: The authors declare that no competing interests exist.

## Introduction

mRNA decay pathways have evolved to both maintain mRNA quality control and to regulate gene expression. For example, the nonsense-mediated mRNA decay (NMD) pathway was initially described for its role in targeting the destruction of messages harboring premature termination codons (PTCs; [*Maquat et al., 1981*; *Peltz et al., 1993*]). The introduction of a PTC can result from genetic mutations or errors in gene expression, ultimately generating a message encoding a potentially deleterious truncated protein. As such, the NMD pathway protects the integrity of the proteome by detecting and destroying PTC-containing messages. In addition to this well-characterized role in quality control surveillance, NMD regulates the expression of ~5–10% of the mammalian transcriptome, highlighting its importance in determining levels of non-aberrant mRNAs (*Mendell et al., 2004*; *Rehwinkel et al., 2005*; *Tani et al., 2012*; *Weischenfeldt et al., 2008*). These apparently normal mRNAs often contain NMD-triggering features such as upstream open-reading frames (uORFs) in the 5'-UTR, the presence of a long 3'-UTR, or a 3'-UTR that harbors an intron (*Karousis et al., 2016*; *Rebbapragada and Lykke-Andersen, 2009*).

NMD is conducted by a conserved set of factors primarily identified via genetic screens in model organisms. The UPF1, 2, and 3 proteins (up-frameshift), discovered in budding yeast, are required for NMD throughout eukaryotes, while the SMG (suppressor with morphogenetic effects on genitalia) proteins were originally identified in *C.elegans* and provide additional enzymatic and regulatory activities required for NMD in metazoans (*Leeds et al., 1991*; *Pulak and Anderson, 1993*); for a recent comprehensive review, see *Karousis et al. (2016)*. UPF1, the central hub of the NMD pathway, is an ATP-dependent RNA helicase that acts at multiple steps in target discrimination and

**eLife digest** The DNA in our cells contains the hereditary information that makes each of us unique. Molecules called mRNAs are copies of this information and are used as templates for making proteins. When a strand of incorrectly copied mRNA, or one including errors from the original DNA template, is recognized, our cells destroy the mRNA to prevent it from producing a damaged protein. Organisms from yeast to humans have evolved a mechanism for finding and destroying faulty mRNAs, called mRNA surveillance. Animals are particularly reliant on mRNA surveillance, as their proteins are often made from cutting and pasting together mRNA from different portions of DNA, in a process known as splicing. Despite being a vital process, we still lack a good understanding of how mRNA surveillance works.

Now, Baird et al. used human kidney cells that produced an error-containing mRNA that could be tracked. To investigate how efficient RNA surveillance is under different conditions, the levels of individual proteins were reduced one at a time. By tracking the amount of faulty mRNA, it was possible to find out if a single protein plays a role in human mRNA surveillance. If the number of faulty mRNAs is high when a protein is reduced, it suggests that this protein may be involved in mRNA surveillance.

Baird et al. screened more than 21,000 proteins, the majority of proteins made in human cells. Many of the proteins that stood out as important in mRNA surveillance were the ones already known to be relevant in yeast and worm cells. But the experiments also identified new proteins that appear to play a role specifically in human RNA surveillance. One of the proteins, ICE1, is essential for the relationship between mRNA splicing and mRNA surveillance. Without ICE1, the mRNA surveillance machinery can no longer find and destroy faulty mRNAs.

Nearly one-third of genetic diseases are caused by mutations that result in faulty mRNAs, which can be detected by mRNA surveillance pathways. Depending on the disease, destroying these error-containing mRNAs can either improve or worsen disease symptoms. A better understanding of the factors that control human RNA surveillance could one day help to develop treatments that affect mRNA surveillance to improve disease outcomes.

DOI: https://doi.org/10.7554/eLife.33178.002

decay. UPF1's ATPase activity and phosphorylation is enhanced by UPF2 binding, promoting decay (*Chakrabarti et al., 2011*; *Chamieh et al., 2008*; *Kashima et al., 2006*). UPF1 phosphorylation by the PI3K-related kinase SMG1 in the context of translation termination promotes the recruitment of the NMD-specific endonuclease SMG6, as well as the recruitment of the CCR4-NOT deadenylase complex via the SMG5-SMG7 heterodimer (*Eberle et al., 2009*; *Gatfield and Izaurralde, 2004*; *Huntzinger et al., 2008*; *Kashima et al., 2006*; *Loh et al., 2013*). UPF1-dependent RNA degradation can thus proceed through a combination of exo- and endonucleolytic pathways.

A second major elaboration of the NMD pathway in vertebrates is the use of the exon junction complex (EJC) to identify potential targets of NMD. The EJC is a tetrameric complex comprising the RNA helicase eIF4AIII, the MAGOH-Y14 heterodimer, and CASC3 (also known as MLN51 and Barentz) and is deposited 20–24 nt upstream of exon-exon junctions during splicing (*Le Hir et al., 2000*). The stable core participates in multiple stages of mRNA function by engaging in dynamic interactions with 'peripheral' EJC factors (*Le Hir et al., 2016*; *Singh et al., 2012*), including UPF3B, which binds EJCs consisting of at least eIF4AIII and MAGOH-Y14 through a conserved motif in its C-terminus (*Buchwald et al., 2010*; *Gehring et al., 2003*). Once bound in the nucleus, the EJC is believed to escort UPF3B into the cytoplasm, where the complex marks most splice sites prior to translation (*Hauer et al., 2016*; *Saulière et al., 2012*; *Singh et al., 2012*). During translation, scanning and elongating ribosomes displace EJCs in the 5'-leader and coding sequence, respectively (*Gehring et al., 2009a*), leaving only EJC-UPF3B complexes > 50 nts downstream of the termination codon bound to the mRNA. UPF3B thus provides a link between nuclear mRNA biogenesis and the cytoplasmic NMD machinery, as the N-terminal region of UPF3B binds UPF2, which in turn promotes UPF1 activity and phosphorylation (*Chamieh et al., 2008*; *Kashima et al., 2006*). Such mRNAs harboring an EJC sufficiently downstream of a stop codon are subject to 'EJC-enhanced' NMD, the branch of the NMD pathway responsible for the most prominent UPF1-dependent decay activities

observed in mammalian cells (*Bühler et al., 2006*; *Cheng et al., 1994*; *Metze et al., 2013*; *Singh et al., 2008*; *Wang et al., 2002*; *Zhang et al., 1998*).

Notably, the primary genetic screens responsible for NMD factor identification were conducted in organisms that do not use the EJC to demarcate targets of decay. Further indicating a potential need for an expanded machinery, the human NMD pathway surveils a transcriptome more complex than that found in yeast and worms and has adopted regulatory roles in development and stress responses (*Feng et al., 2017*; *Gong et al., 2009*; *Karam et al., 2015*; *Medghalchi et al., 2001*; *Wittkopp et al., 2009*). To search for potential novel human NMD proteins, including those involved in 'EJC-enhanced' decay, we used a luciferase reporter to develop a gain-of-signal assay designed based on a well-characterized mRNA β-globin nonsense allele and performed a whole genome RNAi screen (*Maquat et al., 1981*; *Thorne et al., 2010*). By depleting >21,000 genes with three siRNAs/ gene, we recovered core NMD and EJC factors as top hits, as well as a large cohort of potential NMD factors. From the latter, we used biochemical, genetic, and genomic approaches to validate ICE1 (interactor of little elongation complex ELL subunit 1, also known as KIAA0947) as a novel peripheral EJC factor essential for EJC-enhanced NMD. We show that ICE1 depletion leads to enhanced expression of many NMD targets, including those containing PTCs. Further, our data suggest that ICE1 uses a putative MIF4G domain to interact with mature EJCs and is required for proper association of UPF3B with the EJC core proteins. Importantly, overexpressing UPF3B to restore EJC-UPF3B assembly rescues the effects of ICE1 depletion on NMD, consistent with a model in which ICE1 serves an unanticipated role in linking nuclear EJC assembly to cytoplasmic detection of NMD substrates.

## Results

RNAi screens suffer from high rates of false positives arising from sequence-based off-target effects and false negatives due to incomplete protein depletion. Beyond these issues, multiple sources of experimental variance can complicate the identification of true positives. To limit confounding factors inherent to siRNA screens, we designed a bioluminescent gain-of-signal output to identify NMD-associated genes. We chose a luminescence-based reporter system over other assay formats such as fluorescence because luminescence-based assays have superior sensitivity with little or no non-basal background, a significant concern for high-throughput screening (*Fan and Wood, 2007*). In addition, similar assays using *Renilla* luciferase have proven to be sensitive and reliable for mechanistic studies of NMD (*Boelz et al., 2006*; *Holbrook et al., 2006*; *Isken et al., 2008*; *Woeller et al., 2008*). Importantly, our assay strategy results in an increase in bioluminescence with NMD inhibition, thus removing from consideration silenced genes affecting cell viability or gene expression (*Hasson et al., 2015*; *Kaelin, 2017*).

To develop the assay used for high-throughput RNAi screening, we constructed a pair of CMV promoter-driven reporters, a control construct encoding 3XFLAG-tagged firefly luciferase fused in-frame with the wild-type β-globin ORF and an NMD-sensitive construct in which a premature termination codon was introduced at β-globin position 39 (*Figure 1A*). Silencing of critical NMD components by siRNA was expected to inhibit decay of the PTC-containing mRNA, boosting luciferase expression. To confirm that this construct is indeed controlled by the NMD machinery, we co-transfected the plasmids with three UPF1 siRNAs into HEK-293 cells. After a 72 hr incubation, luciferase substrate was added and the luminescence signal was detected. As expected, UPF1 depletion caused a 2- to 3.5-fold increase in luciferase activity from the NMD reporter, relative to the non-silencing siRNA control (*Figure 1B*). This NMD construct expressed a significant basal level of luciferase activity, potentially allowing identification of factors that inhibit the NMD pathway as well as those required for its proper function. Clonal HEK-293 cell lines stably expressing this construct were prepared and used in subsequent characterization and siRNA library screening. To verify the proper functioning of the assay in this context, an siRNA targeting UPF1 was used as a positive control, exhibiting assay dynamic range suitable for high-throughput screening (z'-factors of >0.5).

### Genome-wide RNAi identifies known and putative novel NMD factors

Genome-wide siRNA screening was conducted using a library of siRNAs targeting ~21,000 human genes with three independent siRNAs per gene (*Figure 1C* and *Supplementary File 1*). A major confounding factor in analysis of siRNA screen data is the prevalence of off-target effects mediated

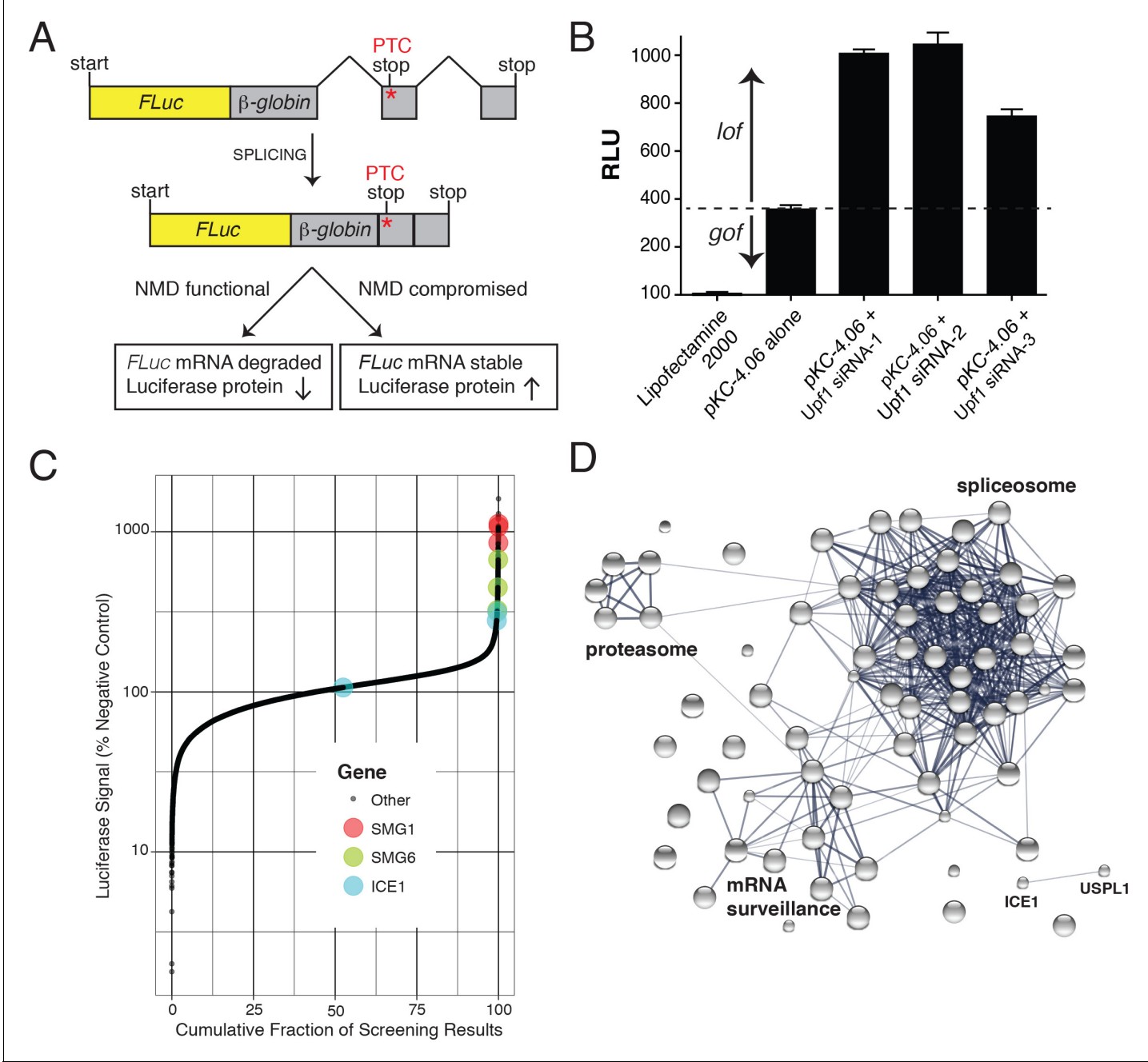

**Figure 1.** FLuc NMD-targeted assay for high-throughput RNAi screening. (**A**) Fusion construct design of the NMD loss-of-function (lof) gain-of-signal (gos) reporter system (3XFLAG-FLuc-βglobinUGA) where a premature termination codon (PTC) within the second exon of the β-globin gene targets the mRNA for rapid NMD (FLuc, firefly luciferase). (**B**) Silencing of the NMD factor UPF1 using three different siRNAs increased expression of transiently expressed luciferase from the NMD 3XFLAG-FLuc-βglobinUGA construct. (**C**) Control normalized, log-transformed MAD Z-scores of genome-wide siRNA screen illustrating high-ranking genes from the NMD pathway. (**D**) STRING pathway analysis illustrating tight connectivity of highest ranking genes into RNA-associated processing pathways.

DOI: https://doi.org/10.7554/eLife.33178.003

The following figure supplements are available for figure 1:

**Figure supplement 1.** STRING network analysis of genes identified in RNAi screen.
DOI: https://doi.org/10.7554/eLife.33178.004

**Figure supplement 2.** Comparison of siRNA and CRISPR screens.
DOI: https://doi.org/10.7554/eLife.33178.005

**Figure supplement 3.** Phylogenetic alignment of the predicted ICE1 C-terminal MIF4G domain.

*Figure 1 continued on next page*

*Figure 1 continued*

DOI: https://doi.org/10.7554/eLife.33178.006

**Figure supplement 4.** Predicted MIF4G domain of ICE1 modeled on third MIF4G domain of UPF2.

DOI: https://doi.org/10.7554/eLife.33178.007

by siRNA seed sequences. The large number of siRNAs used in this screen enables application of Common Seed Analysis (CSA), a computational approach to adjust for seed sequence-based off-target effects (*Marine et al., 2012*). Following CSA, we ranked genes by median seed-corrected Z-score of the corresponding siRNAs and subjected the top ranked hits (seed corrected Z-score >1.5, 76 genes) to pathway analysis using STRING (*Figure 1D*, *Table 1*, and *Figure 1—figure supplement 1*; [*Szklarczyk et al., 2017*]). Several canonical members of the NMD pathway were identified with seed-corrected Z-scores > 1.5, including UPF1, SMG1, SMG5, SMG6, and SMG7. All four members of the core EJC also met this stringent cutoff (RBM8A, EIF4A3, MAGOH, and CASC3), along with DHX8/PRP22 and Aquarius (AQR), proteins implicated in EJC deposition (*Ideue et al., 2007*). NMD pathway components UPF2 and UPF3B also exhibited a positive response in the screen, with seed corrected Z-scores of 0.98 and 0.72, respectively. In addition to proteins known to directly participate in NMD, spliceosomal proteins were highly enriched among genes whose depletion caused elevated luciferase expression, with particular over-representation of proteins from the PRP19-related, Sm, and U2 snRNP sub-complexes (*Supplementary file 2*). Overall, the strong enrichment of genes involved in mRNA processing and NMD indicates that the expected biology was identified by the screen and that other high-scoring genes are likely to be members of the same processes. Moreover, the performance of the screen compares favorably to a recent CRISPR-based screen for NMD factors (*Figure 1—figure supplement 2*), particularly with respect to identification of EJC and related

**Table 1.** Known NMD factors are enriched among top hits of siRNA screen.

Whole-genome siRNA screen median seed corrected Z-scores. Wilcoxon test p-values, and rank Z-scores for selected genes known to be involved in NMD are shown.

|  | Gene | Median seed corrected Z-score | p-Value | Rank |
|---|---|---|---|---|
| UPF Proteins | UPF1 | 1.76 | 3.05E-03 | 62 |
|  | UPF2 | 0.98 | >0.05 | 193 |
|  | UPF3B | 0.72 | 1.15E-02 | 470 |
| SMG Proteins | SMG1 | 7.60 | 2.41E-03 | 1 |
|  | SMG5 | 2.25 | 3.20E-03 | 47 |
|  | SMG6 | 4.75 | 2.47E-03 | 14 |
|  | SMG7 | 2.14 | 2.77E-03 | 51 |
|  | SMG8 | 0.44 | 2.28E-03 | 1469 |
|  | SMG9 | 0.17 | >0.1 | 3859 |
| EJC Proteins | EIF4A3 | 6.90 | 2.41E-03 | 3 |
|  | RBM8A | 7.04 | 2.41E-03 | 2 |
|  | MAGOH | 5.65 | 2.43E-03 | 8 |
|  | CASC3 | 3.92 | 2.57E-03 | 19 |
| EJC Associated | CWC22 | 1.07 | 1.22E-02 | 150 |
|  | AQR | 6.46 | 2.42E-03 | 6 |
|  | ACIN1 | 0 | >0.1 | 13890 |
|  | RNPS1 | −0.70 | >0.1 | 20388 |
| Other | NBAS | 0.26 | >0.1 | 2955 |
|  | PNRC2 | 0.52 | >0.05 | 1074 |
|  | DHX34 | 0 | >0.1 | 6499 |

DOI: https://doi.org/10.7554/eLife.33178.008

proteins. Despite the fact that both screens successfully enriched for proteins known to be involved in NMD, only 13 genes were shared as significant hits between the two screens, five of which were CASC3, SMG5, SMG6, SMG7, and UPF1.

Following the whole-genome screen, 135 genes were further interrogated with additional siRNAs (*Supplementary file 3*). Of these, 27 exhibited seed-corrected Z-scores greater than 1, including known pathway components SMG1, SMG6, SMG7, EIF4A3, RBM8A, MAGOH, CASC3, CWC22, and AQR. Among the genes whose depletion caused reproducible effects on luciferase expression using multiple siRNAs, we manually curated candidates for downstream analysis, focusing on those with features suggestive of a possible role in NMD. Two promising examples were known interacting partners ICE1/KIAA0947 and the SUMO isopeptidase USPL1 (primary screen seed-corrected Z-scores of 2.91 and 3.01, respectively). These proteins are known to associate with the little elongation complex, a protein assembly that interacts with and promotes RNA polymerase II association and elongation at snRNA promoters (*Hu et al., 2013*). ICE1 is a ~ 250 kDa nuclear protein proposed to have a scaffolding role in little elongation complex assembly but has not been structurally characterized or previously implicated in NMD (*Hu et al., 2013*; *Hutten et al., 2014*). To gain insight into possible functions of ICE1 in NMD, we performed protein structure prediction using the Phyre2 web portal (*Kelley et al., 2015*). This analysis identified a putative MIF4G domain with high confidence at the C-terminus of the protein, a region highly conserved among vertebrates (*Figure 1—figure supplement 3* and *Figure 1—figure supplement 4*). MIF4G domains are prevalent in proteins involved in translation and NMD (*Barbosa et al., 2012*; *Buchwald et al., 2013*; *Kadlec et al., 2004*), leading us to hypothesize that ICE1 may use its putative MIF4G domain to interact with components of the NMD pathway.

## ICE1 depletion increases abundance of transcripts with NMD-inducing features

To assess a possible role for ICE1 in regulating NMD, we performed genome-wide RNAseq analyses of cells transfected with control (siNT), ICE1, and UPF1 siRNAs (*Supplementary file 4*). We began by evaluating the effect of ICE1 depletion on classes of transcripts with known NMD-inducing features. As the luciferase-β-globin reporter used to identify ICE1 undergoes EJC-stimulated decay, we first examined the impact of ICE1 depletion on transcript isoforms containing stop codons at least 50 nt upstream of the final exon-exon junction (PTCs). For these analyses, we focused on genes for which at least one alternative PTC-containing isoform and one normal isoform were represented in our RNAseq data, finding that ICE1 depletion specifically increased the abundance of the PTC-containing isoforms relative to the normal isoforms (*Figure 2A*). These data provide initial evidence that ICE1 is important for cellular elimination of many canonical NMD targets.

To further investigate a role for ICE1 in EJC-dependent NMD, we tested its involvement in regulating the SR protein SRSF2 (also known as SC35). SRSF2 represents a classic example of NMD-based autoregulation of proteins involved in alternative splicing. Accumulation of SRSF2 protein leads to excision of two introns from its own 3'UTR, generating an NMD target mRNA referred to here as SRSF2(NMD+) ([*Sureau et al., 2001*]; *Figure 2B*, top schematic). Using previously characterized primers in the SRSF2 3'-UTR to distinguish SRSF2 transcript isoforms ([*Hauer et al., 2016*]; arrows denote region of amplification), our results recapitulate previous findings that SRSF2(NMD+) is a potent NMD substrate, as depletion of UPF1 resulted in a seven-fold increase in the NMD+ isoform compared to non-targeting controls (*Figure 2B*, bottom right bar graph). Consistent with our RNAseq studies, depletion of ICE1 resulted in a four-fold increase in the abundance of this NMD target (*Figure 2B*, bottom right graph), while levels of the unspliced 3'UTR isoform SRSF2(NMD-) were not increased. These findings indicate that increases in the SRSF2(NMD+) isoform are due to disruption of EJC-stimulated NMD rather than increased transcription from the SRSF2 locus.

Transcripts containing uORFs represent a second prominent class of NMD targets (*Gardner, 2008*; *Hurt et al., 2013*). If uORFs are efficiently used relative to the major transcript ORF, the uORF TC can be recognized as a premature termination codon and degraded by NMD. We examined genes represented in a translation initiation site (TIS) database based on ribosome profiling data from HEK-293 cells (*Wan and Qian, 2014*). Of the 4310 genes expressed in our RNAseq data that also met ribosome profiling read coverage cutoffs for inclusion in the TIS database, 1578 showed evidence of uORF translation. As expected of a putative NMD factor, depletion of ICE1 systematically led to increased abundance of mRNAs with empirically identified uORFs, compared to

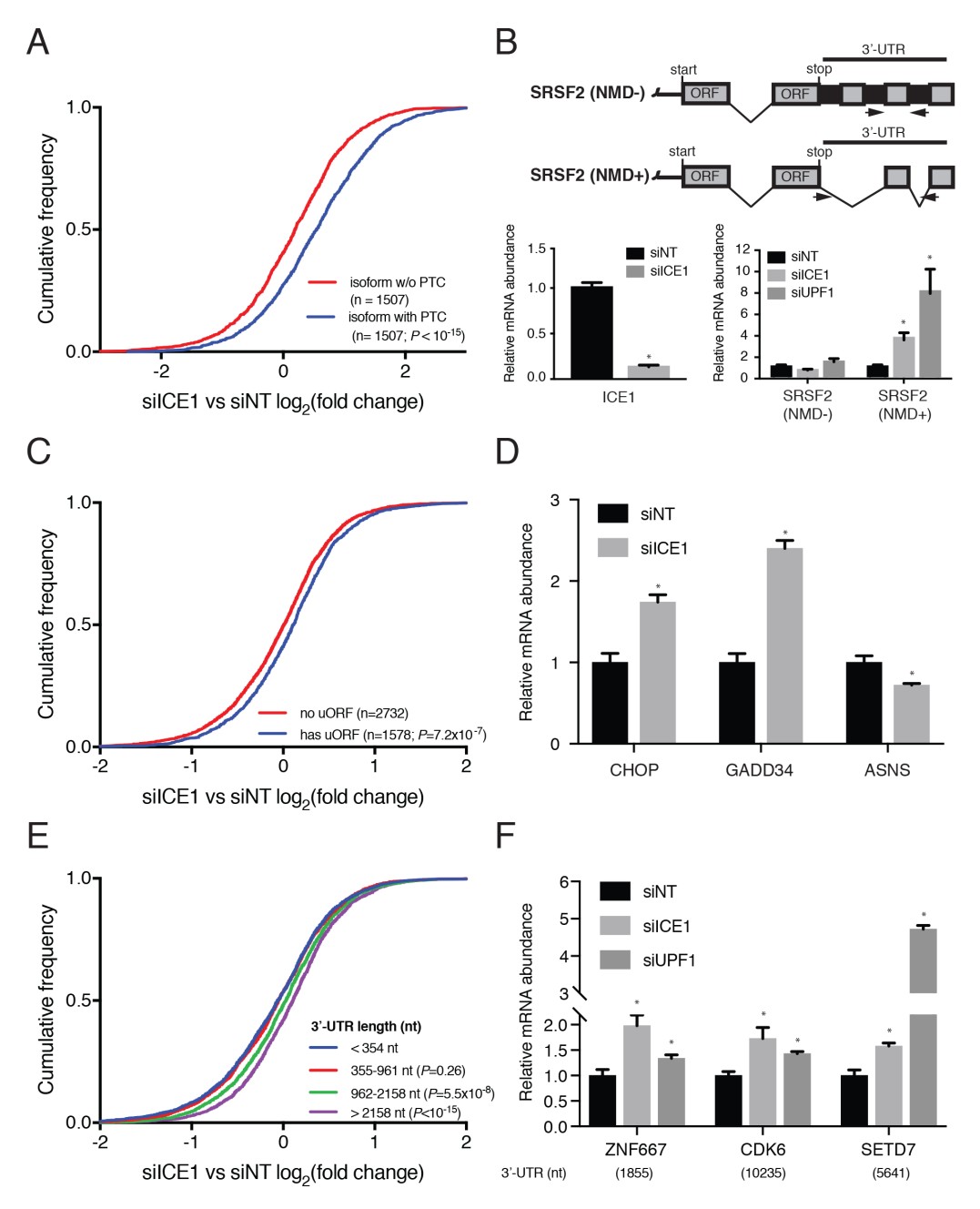

**Figure 2.** ICE1 depletion results in increased abundance of NMD substrates. (**A**) CDF plot of the effect of ICE1 depletion on mRNA isoforms containing and lacking PTCs as defined using the 50 nt rule. Only genes with detectable expression of both isoform types were considered. Statistical significance was determined by K-S test. (**B**) (Top) Schematic illustrating SRSF2 splicing isoforms with differential sensitivity to the NMD pathway. (Bottom) RT-qPCR analysis of NMD-insensitive and -sensitive SRSF2 transcript isoforms during ICE1 and UPF1 depletion. ICE1 knockdown efficiency shown on the left is for the cDNA libraries used in *Figure 2B,D and F*. (**C**) CDF plot of the effect of ICE1 depletion on expression of mRNAs previously determined to contain or lack actively translated uORFs. Statistical significance was determined by K-S test. (**D**) RT-qPCR analysis of uORF-containing mRNAs CHOP and GADD34 during ICE1 depletion. (**E**) Response to ICE1 depletion among genes (n = 11057) divided into quartiles according to the 3'UTR length of the most highly expressed isoform detected in Kallisto analyses. Statistical significance was determined by K-S test. (**F**) RT-qPCR analysis of NMD substrates with long 3'-UTRs during ICE1 and UPF1 depletion. 3'-UTR lengths are provided in nucleotides (nt) beneath the X-axis. Data represent mean values with error bars illustrating standard deviation. Asterisks indicate statistical significance (n = 3, Student's t-test, p<0.05) between non-targeting and gene-specific siRNA knockdown values.

DOI: https://doi.org/10.7554/eLife.33178.009

The following figure supplements are available for figure 2:

*Figure 2 continued on next page*

*Figure 2 continued*

**Figure supplement 1.** Analysis of the effect of ICE1 and UPF1 depletion on mRNAs with uORFs and long 3'UTRs.
DOI: https://doi.org/10.7554/eLife.33178.010
**Figure supplement 2.** RNAseq traces for NMD targets during control and ICE1 depletion.
DOI: https://doi.org/10.7554/eLife.33178.011

those lacking evidence of uORF translation (*Figure 2C*). The extent of up-regulation among uORF-containing mRNAs was less than that observed for PTC-containing mRNAs (*Figure 2A*), possibly because leaky scanning or efficient re-initiation at downstream ORFs is sufficient to stabilize many mRNAs undergoing some degree of uORF translation (*Neu-Yilik et al., 2011*; *Zhang and Maquat, 1997*). Because ICE1 participates in snRNA biogenesis as part of the little elongation complex, we asked whether the observed increases in uORF-containing mRNAs could be attributed to alterations in splicing. We detected alternative splicing events using two independent software packages, Majiq and Leafcutter, and excluded any genes found to undergo of splicing changes upon ICE1 depletion by either algorithm, using permissive cutoffs (*Supplementary file 5*; see Materials and methods for details; [*Li et al., 2018*; *Vaquero-Garcia et al., 2016*]). Removing these genes from the analysis of uORF-containing mRNAs had no apparent impact (*Figure 2—figure supplement 1A*). Likewise, removing the genes for which evidence of a PTC-containing isoform was detected (*Figure 2A*) had no effect on the relationship between ICE1 depletion and uORF mRNA abundance (*Figure 2—figure supplement 1B*).

Despite advances in uORF identification, there remain few human transcripts in which relative translation efficiencies of uORFs and primary ORFs are well characterized. For this reason, we chose to validate ICE1's role in uORF-directed decay by studying key regulators of the Integrated Stress Response (ISR) known to be subject to translational regulation through uORFs that are inhibitory to translation of the downstream codon sequence (CDS). Under normal conditions, expression of both the transcription factor CHOP/DDIT3 and the phosphatase regulatory subunit GADD34/PPP1R15A are impaired by a mechanism involving translation of an inhibitory uORF that prevents downstream reinitiation at the CDS (*Young and Wek, 2016*). The inability of the ribosome to translate the CDS and displace UPF1 and/or EJCs predisposes these transcripts for NMD (*Karam et al., 2015*; *Schmidt et al., 2015*; *Weischenfeldt et al., 2008*). To investigate the effect of ICE1 depletion on transcript abundance of the ISR members CHOP/DDIT3 and GADD34/PPP1R15A, we performed RT-qPCR on RNA from cells treated with siICE1 or a non-targeting control siRNA. During ICE1 depletion, there was a 2–2.5 fold increase in the abundance of these uORF-containing transcripts (*Figure 2D*). To ensure that this was not an artifact of ICE1 depletion potentially inducing the ISR, we also measured levels of the downstream ISR member ASNS, which is transcriptionally induced during the ISR but not under translational control (*Baird et al., 2014*; *Barbosa-Tessmann et al., 2000*). ICE1 depletion did not increase ASNS levels (*Figure 2D*), and we did not observe alterations in mRNA processing of these and other genes induced by ICE1 depletion (see below; *Figure 2—figure supplement 2*), suggesting that the increase in CHOP/DDIT3 and GADD34/PPP1R15A levels was the product of reduced NMD activity.

Due to the large number of human NMD targets made susceptible to decay via long 3'UTRs, we also examined the impact of ICE1 depletion on abundance of transcripts with varying 3'UTR lengths. We stratified genes into quartiles on the basis of the 3'UTR length of their most highly expressed isoforms in siNT-treated cells. The responses of genes with 3'UTRs in the shortest two quartiles (<354 nt and 355–961 nt) were indistinguishable, but ICE1 depletion resulted in significantly increased abundance of genes with 3'UTRs in the longest two quartiles (962–2158 nt and >2158 nt; *Figure 2E*). The failure of ICE1 depletion to enhance abundance of mRNAs with 3'UTRs of moderate length (355–961 nt) differs from the significantly enhanced accumulation of such mRNAs in UPF1-depleted cells, but further work will be required to determine whether this is due to biological or technical differences (*Figure 2—figure supplement 1C*). As with the uORF-containing mRNAs, exclusion of mRNAs exhibiting ICE1-dependent splicing changes or expression of PTC-containing isoforms had no effect on the analysis of ICE1 depletion and 3'UTR-dependent changes in mRNA abundance (*Figure 2—figure supplement 1D and E* and *Supplementary file 5*). In subsequent qRT-PCR studies, ICE1 knockdown led to increased abundance of some long 3'UTR-containing mRNAs sensitive to UPF1 depletion, including ZNF667 (1855 nt 3'UTR), CDK6 (10,235 nt), and

SETD7 (5461 nt) (*Figure 2F* and *Figure 2—figure supplement 2*). Together, our RNAseq analyses suggest that ICE1 depletion can disrupt NMD induced by uORFs, PTCs, and long 3'UTRs.

## ICE1 depletion stabilizes mRNAs with NMD-inducing features

To investigate whether the observed changes in mRNA abundance were due to increases in mRNA stability consistent with NMD inhibition, we used REMBRANDTS software to infer changes in mRNA stability from the relative abundance of exonic and intronic reads from each gene (*Alkallas et al., 2017*). The REMBRANDTS algorithm is a refinement of earlier methods based on the idea that a change in exonic reads without a corresponding change in intronic reads is diagnostic of differential RNA stability, while concurrent changes in both exonic and intronic reads suggest altered transcription (*Ameur et al., 2011*; *Gaidatzis et al., 2015*; *Zeisel et al., 2011*). The effects of ICE1 depletion on mRNA abundance and relative stability were highly correlated (Spearman's ρ=0.69; p<10$^{-15}$; *Figure 3—figure supplement 1A* and *Supplementary file 6*), and transcripts that significantly increased in steady-state abundance with ICE1 knockdown were preferentially stabilized upon ICE1 or UPF1 depletion (*Figure 3A* and *Figure 3—figure supplement 1B*). Likewise, mRNAs that were increased in abundance upon UPF1 knockdown tended to be stabilized by ICE1 knockdown or UPF1 knockdown (*Figure 3—figure supplement 1C and D*). Comparison of relative mRNA stability upon ICE1 depletion with responses to UPF1 or UPF3B siRNAs revealed that the populations of mRNAs stabilized by depletion of these two core NMD factors were stabilized to a similar extent by ICE1 depletion (*Figure 3B*). Moreover, the mRNAs stabilized in both UPF1 and UPF3B knockdown cells exhibited an enhanced response to ICE1 depletion. Finally, we asked whether ICE1 depletion preferentially stabilizes mRNAs with uORFs or long 3'UTRs, with results that closely mirrored our findings based on steady-state mRNA abundance measurements (compare *Figure 2C and E* with *Figure 3—figure supplement 1E and F*).

To independently assess whether ICE1 affects NMD target mRNA stability, we used a metabolic labeling approach to quantify mRNA half-lives in cells transfected with non-silencing control, UPF1, or ICE1 siRNAs (*Dölken, 2013*; *Russo et al., 2017*). Following gene depletion, cells were pulse-labeled with 5-ethynyluridine, a modified nucleoside that allows covalent biotinylation and quantitative recovery of newly transcribed mRNA. Transcript half-lives can then be calculated based on the relative abundance of nascent and total mRNA. As observed in our transcriptome-wide analyses, these metabolic labeling studies indicated that either ICE1 or UPF1 depletion increased the stability of several well-characterized NMD target mRNAs, while failing to stabilize the non-NMD target control ASNS (*Figure 3C*; *Figure 3—figure supplement 2*). Together, these results indicate that the effect of ICE1 depletion on steady-state levels of mRNAs with NMD-inducing features is due to increased mRNA stability, consistent with a role for the protein in promoting NMD through UPF1 and UPF3B.

## ICE1 depletion causes concomitant increases in mRNA and protein expression

NMD execution depends on mRNA nuclear export and translation in the cytoplasm, meaning that interference with multiple aspects of mRNA biogenesis, transport, and function can cause indirect inhibition of decay. The gain of luciferase signal observed with ICE1 siRNAs in the whole-genome screen suggests that ICE1 depletion did not prevent efficient NMD reporter export and translation (*Figure 1C* and *Supplementary file 1*). To corroborate this finding using endogenous NMD substrates, we detected several proteins produced from UPF1- and ICE1-sensitive mRNAs by immunoblotting (*Figure 3D*). In all cases, we observed substantial increases in protein production from the target mRNAs, indicating that ICE1 depletion increases stability of NMD target mRNAs without interfering with their translation in the cytoplasm.

## ICE1 associates with the core EJC

Based on our observations implicating ICE1 in NMD, we next tested whether ICE1 associates with NMD or EJC proteins. Whole cell lysates from HEK-293 cells were subjected to immunoprecipitation assays using antibodies against ICE1, with UPF3B antibodies as a positive control for recovery of both the trimeric UPF complex and EJC components. Immunoblot analysis of samples purified with these specific antibodies or control IgGs revealed ICE1 co-purification with the core EJC component

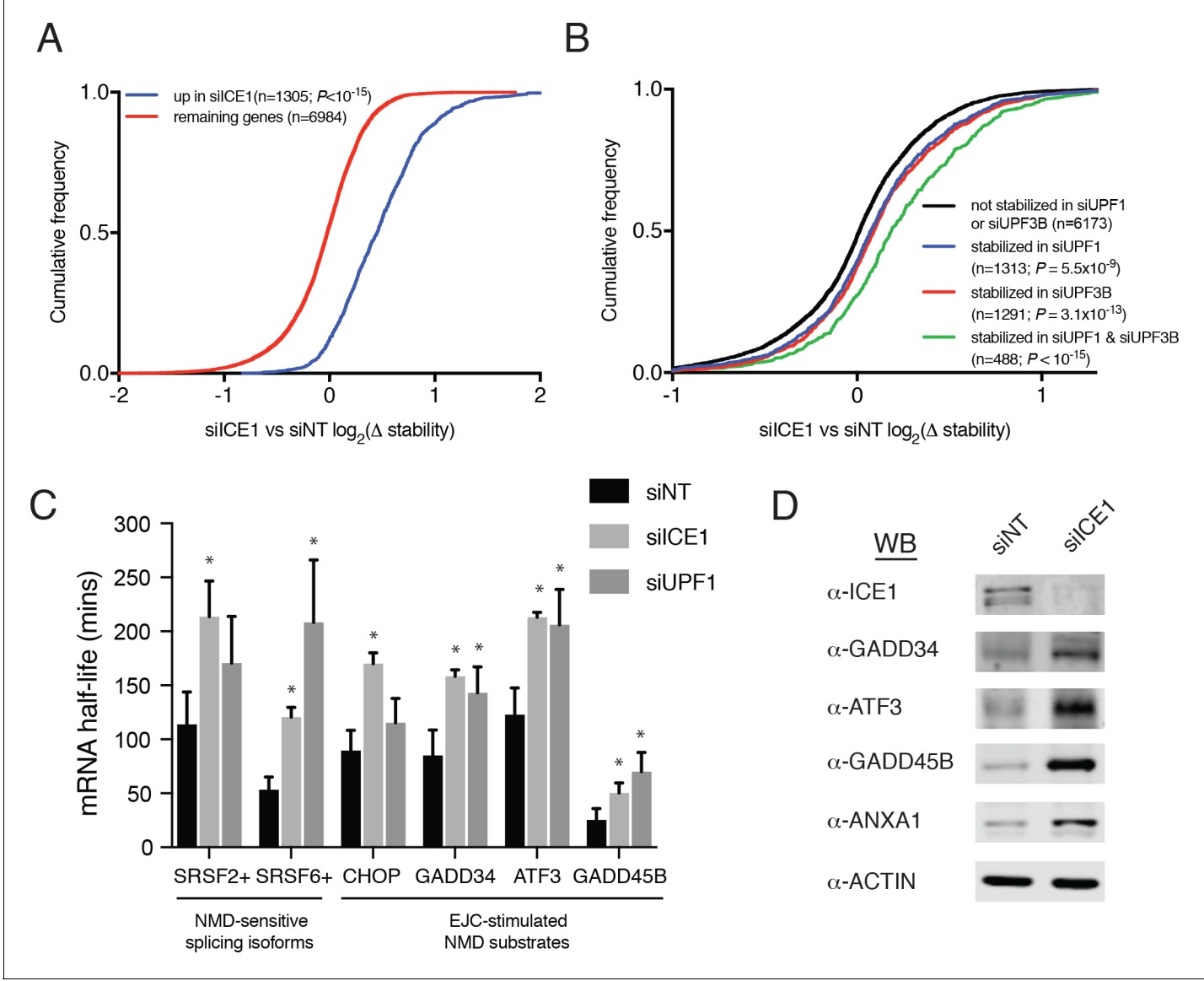

**Figure 3.** ICE1 depletion causes stabilization and translation of NMD target mRNAs. (**A**) CDF plot of relative mRNA stability as determined by REMBRANDTS analysis of RNAseq following siICE or siNT treatment, categorized according to the change in steady-state mRNA levels upon siICE1 treatment. Transcripts were classified as increased in siICE1 if they exhibited a log2(FC) >0.5 and a Sleuth q-value <0.05. Statistical significance was determined by K-S test. (**B**) CDF plot as in (**A**), with genes categorized according to their relative stability changes in response to UPF1 or UPF3B knockdown. Transcripts with log2(Δstablity)>0.2 and p-value<0.05 were classified as stabilized. Statistical significance was determined by K-S test. (**C**) Gene-specific mRNA half-lives during siRNA depletion as determined by 5-EU metabolic labeling. Bars represent mean half-life values from three biological replicates, error bars indicate standard deviation, and asterisks mark significant differences between treatments (Student's t-test; p<0.05). (**D**) Western blot of protein levels from HEK-293 lysates depleted of ICE1 (siICE1) or a non-targeting control (siNT).
DOI: https://doi.org/10.7554/eLife.33178.012

The following figure supplements are available for figure 3:

**Figure supplement 1.** Analysis of mRNA stability changes upon UPF1 and ICE1 depletion.
DOI: https://doi.org/10.7554/eLife.33178.013

**Figure supplement 2.** ICE1 and UPF1 depletion controls and knockdown efficiencies in metabolic labeling studies.
DOI: https://doi.org/10.7554/eLife.33178.014

eIF4AIII, (*Figure 4A*), but not NMD proteins UPF1, UPF2, or UPF3B. Conversely, UPF3B co-purified with eIF4AIII, UPF1, and UPF2, but not ICE1. The immunoprecipitation protocol used involves hypotonic lysis and gentle buffer conditions to retain RNA-binding protein interactions with RNAs (see Materials and methods). To further investigate the nature of the interaction between ICE1 and eIF4AIII, we determined whether coprecipitation was disrupted by thorough digestion with an RNase A/T1 cocktail. Indicative of RNA-independent complex formation, ICE1 coprecipitation with eIF4AIII was not perturbed by RNase digestion. Importantly, in the UPF3B-positive control

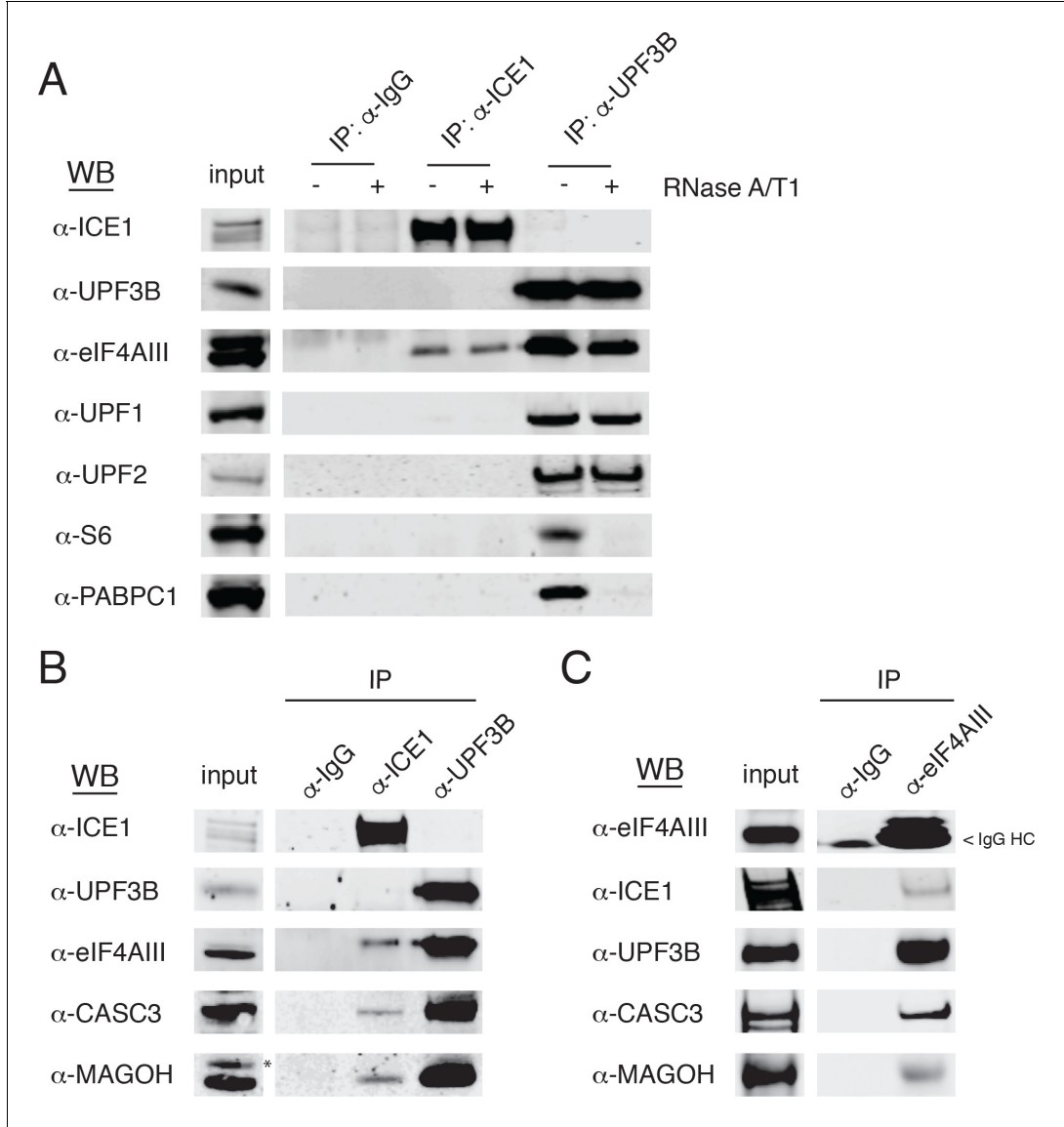

**Figure 4.** ICE1 co-immunopurifies with the core EJC in HEK-293 cells. (**A**) Western blot showing lysates subject to co-immunoprecipitation with control IgG, ICE1 or UPF3B antibodies. Lysates were treated with RNase A/T1 cocktail during the copurification (+) or untreated (-), and antibodies against the indicated proteins were used for detection. (**B**) Western blot of cell lysates subject to co-immunoprecipitation with control IgG, ICE1 or UPF3B antibodies. Antibodies against the core EJC members used for detection are indicated at the left, and asterisk indicates non-specific band. (**C**) Western blot of lysates subject to co-immunoprecipitation with control IgG or monoclonal eIF4AIII antibody. Antibodies used for detection of the core and peripheral EJC components are indicated. Input lanes represent 3% of the total immunoprecipitated material.

DOI: https://doi.org/10.7554/eLife.33178.015

The following figure supplement is available for figure 4:

**Figure supplement 1.** ICE1 copurifies with overexpressed UPF3B in an EJC-dependent manner.

DOI: https://doi.org/10.7554/eLife.33178.016

immunopurifications, RNase treatment abolished recovery of ribosomal protein S6 and PABPC1, suggesting that the RNase conditions used efficiently disrupted RNA-mediated interactions (*Figure 4A*).

We also observed recovery of CASC3 and MAGOH in samples immunopurified with the antibody against ICE1, indicating that ICE1 interacts with fully assembled EJCs (*Figure 4B*). Reciprocally, we used monoclonal antibodies against eIF4AIII to coprecipitate ICE1, along with the expected partners UPF3B, MAGOH, and CASC3 (*Figure 4C*). Of note, we were unable to recover UPF3B or ICE1 from FLAG co-immunoprecipitations when the N- or C-termini of eIF4AIII were tagged, consistent with a previously reported mass spectrometry dataset (data not shown; [*Singh et al., 2012*]). Furthermore, while we did not observe co-purification of ICE1 and UPF3B using antibodies against endogenous proteins, stably overexpressed 3XMYC-UPF3B recovered ICE1, in a manner dependent on the UPF3B EJC-binding domain (*Figure 4—figure supplement 1*). These findings indicate that ICE1 can participate in a EJC-UPF3B complex but preferentially associates with fully assembled EJCs not bound to UPF3B.

## A putative MIF4G domain mediates ICE1-EJC interactions and inhibits NMD when overexpressed

As described above, ICE1 encodes a putative C-terminal MIF4G domain (*Figure 1—figure supplement 4* and *Figure 5—figure supplement 1A*). To determine if the C-terminus of ICE1 is important for EJC interaction, we transiently expressed 3XFLAG-tagged full-length ICE1 (*Figure 5—figure supplement 1A*; 3XF-ICE1), a variant lacking the C-terminus (3XF-ICE1 N-term), or the putative C-terminal MIF4G domain (3XF-MIF4G$^{ICE1}$). The full-length ICE1 protein and the isolated C-terminus recovered endogenous eIF4AIII and CASC3 above background levels, but the protein lacking the C-terminus failed to enrich for the EJC proteins (*Figure 5—figure supplement 1B*). To further probe the interaction between the putative MIF4G domain and the EJC, we employed stable Flp-In 293-TREx cell lines that allow for the integration of a single, inducible copy of a transgene of interest. Using stable expression of 3XF-MIF4G$^{ICE1}$, we again observed that putative ICE1 MIF4G domain was sufficient for association with eIF4AIII (*Figure 5A*). This interaction was retained during extensive RNase digestion, consistent with co-immunopurification studies using antibodies against endogenous proteins (*Figure 4A*).

As the C-terminal ~30 kDa of ICE1 is sufficient for its interaction with EJC proteins, we asked whether overexpression of the putative MIF4G domain could have a dominant negative effect. Consistent with this hypothesis, stable overexpression of the (3XFLAG-MIF4G$^{ICE1}$) protein used for immunoprecipitation experiments caused a reproducible several-fold increase in levels of two mRNAs that responded strongly to both ICE1 and UPF1 knockdown in RNAseq and qRT-PCR experiments, ANXA1 (which contains a putative uORF; [*Thierry-Mieg and Thierry-Mieg, 2006*]) and CGA (previously identified as a strong target of UPF3B-dependent NMD; (*Chan et al., 2007*); *Figure 5B*). We note that this effect is not observed on all NMD targets tested, as the NMD substrate ATF3 was not stabilized during MIF4G overexpression, but is likely only apparent on highly NMD-sensitive transcripts such as ANXA1 and CGA (*Figure 5B*). Together, these data indicate that ICE1's interaction with eIF4AIII occurs through its putative MIF4G domain and that its pro-NMD function is disrupted when this domain is expressed in trans.

## ICE1 is required for the link between nuclear EJC assembly and cytoplasmic decay

Having observed ICE1 co-purification with endogenous core EJC proteins but not UPF3B, we next investigated whether NMD inhibition upon ICE1 depletion was due to disrupted EJC function. To determine whether ICE1 modulates EJC interactions with peripheral factors, we immunopurified fully assembled EJCs and associated proteins using an antibody against CASC3. In lysates from HEK-293 cells depleted of ICE1 (siICE1), CASC3 co-immunoprecipitated with EJC core members eIF4AIII and MAGOH at comparable levels as in the non-targeting (siNT) control (*Figure 6A*, left immunoblot, right bar graph illustrating eIF4AIII/CASC3 recovery efficiency quantified from three independent replicates), suggesting that ICE1 is not required for core EJC assembly. Interestingly, further immunoblotting of proteins co-purified with CASC3 showed a drastic reduction in the ability of the mature EJC to associate with NMD factors UPF3B and UPF2 when ICE1 was depleted. Following ICE1

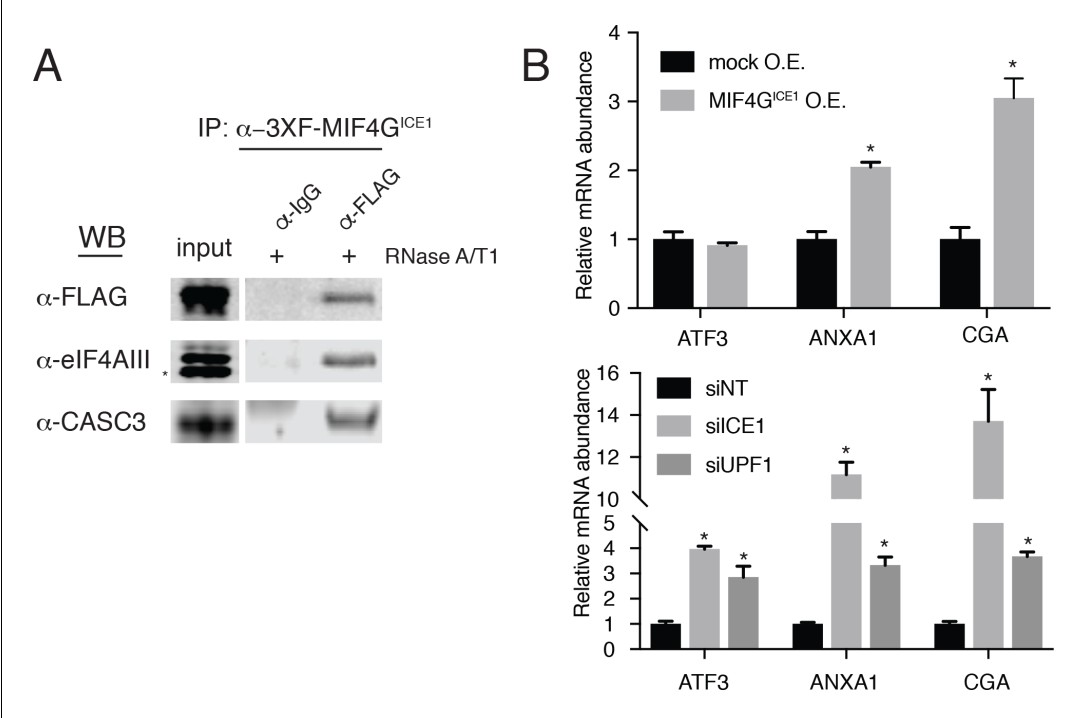

**Figure 5.** The putative MIF4G domain of ICE1 is sufficient to interact with eIF4AIII. (**A**) Western blot of cell lysates stably expressing mock or 3XFLAG-tagged putative ICE1 MIF4G domain (3XFLAG-MIF4G$^{ICE1}$) were subjected to coimmunoprecipitation with a FLAG antibody. Lysates were treated with RNase A/T1 cocktail during the copurification (+), and antibodies used for detection are indicated at the left. Input lanes represent 3% of the total immunoprecipitated material. (**B**) (Top) RT-qPCR analysis of the NMD targets ATF3, ANXA1 and CGA in HEK-293 cells expressing a mock plasmid or vector expressing 3XFLAG-tagged putative ICE1 MIF4G domain. (Bottom) RT-qPCR analysis of ATF3, ANXA1 and CGA levels in cells depleted of ICE1 or UPF1. ICE1 knockdown efficiency is represented in *Figure 2B*. Data represent mean values with error bars illustrating standard deviation. Asterisks indicate statistical significance between control and 'treatment' values for each experiment (n = 3, Student's t-test, p<0.05).
DOI: https://doi.org/10.7554/eLife.33178.017

The following figure supplement is available for figure 5:

**Figure supplement 1.** The C-terminus of ICE1 is sufficient to recover eIF4AIII.
DOI: https://doi.org/10.7554/eLife.33178.018

depletion, CASC3 coprecipitated ~70% less UPF3B and, by association, nearly undetectable levels of UPF2 as compared to the non-targeting control (*Figure 6A*, left immunoblot, right bar graph illustrating UPF3B/CASC3 recovery efficiency quantified from three independent replicates). We observed similar disruption of UPF3B-EJC interactions upon overexpression of the putative ICE1 MIF4G domain, suggesting that the phenotypes arising from these interventions share a mechanistic basis (*Figure 6—figure supplement 1*).

The defect in UPF3B-EJC association in the absence of ICE1 could be due to impaired RNP assembly in the nucleus or failure to retain interactions in the cytoplasm. To begin to explore the role of ICE1 in regulating EJC-UPF3B interactions, we examined the localization of stably expressed GFP-tagged UPF3B in cells treated with control or ICE1 siRNAs. Following siRNA transfection, we marked nuclei with Hoechst stain and subjected cells to imaging flow cytometry (*Figure 6B*). This approach allowed quantitative determination of the nuclear-cytoplasmic distribution of GFP-UPF3B in thousands of cells. In control cells, GFP-UPF3B was predominantly nuclear, but prominent diffuse cytoplasmic signal could also be observed, resulting in a median nuclear:cytoplasmic ratio of 2.89 ± 0.85 S.E.M. In contrast, ICE1 depletion caused UPF3B to accumulate in nuclei and be depleted from the cytoplasm. In knockdown cells, the median nuclear:cytoplasmic ratio shifted significantly to 5.59 ± 2.65 S.E.M. (Mann Whitney test, $p<10^{-15}$), presumably decreasing UPF3B's capacity to interact with mRNA targets of decay. It is thought that UPF3B is exported from the nucleus in association with mRNA-bound EJCs (*Gehring et al., 2009b*). To test whether reduced EJC binding contributes to increased nuclear accumulation of UPF3B, we analyzed the subcellular

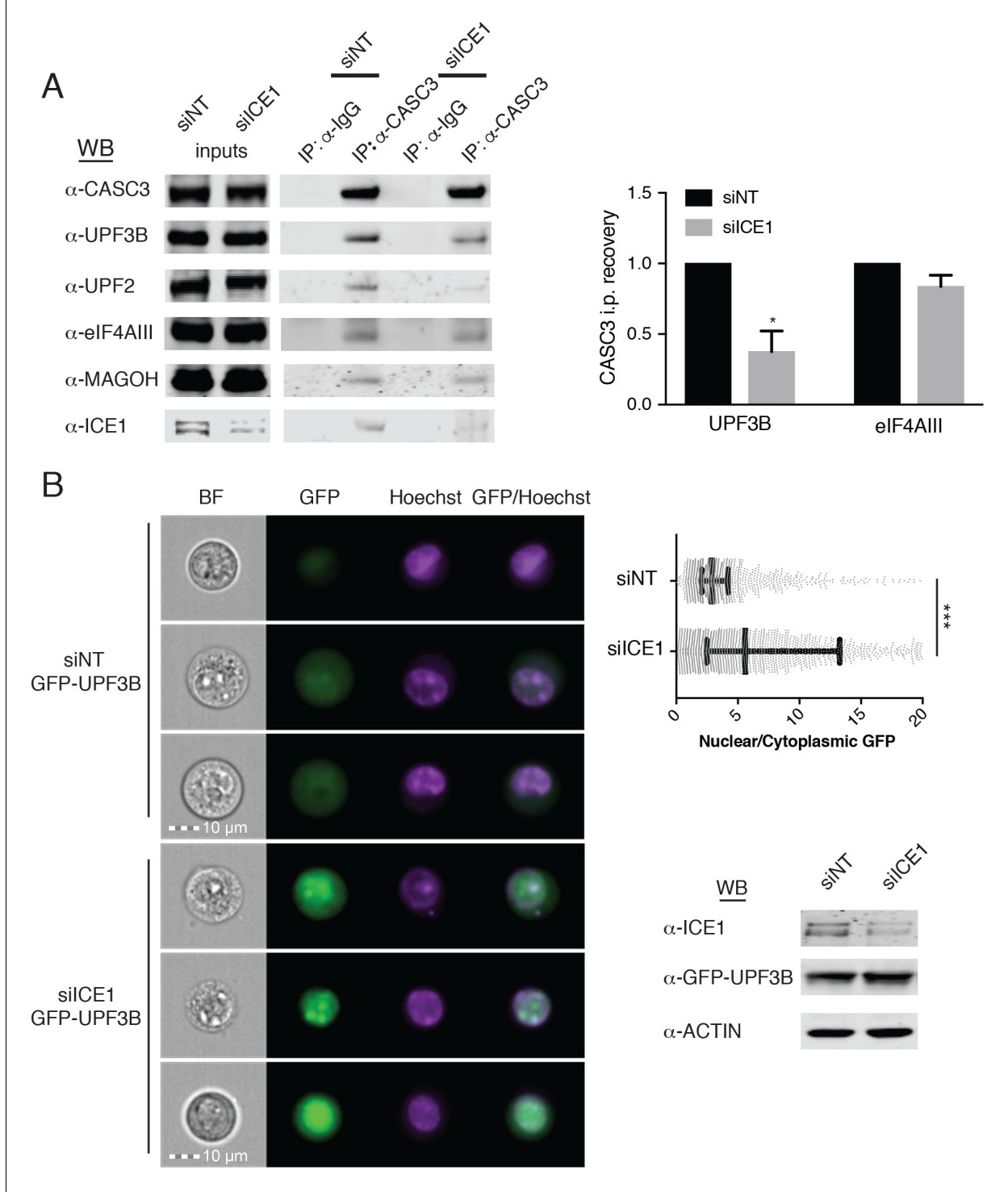

**Figure 6.** ICE1 is required for UPF3B association with the EJC. (**A**) (Left) Western blot of cell lysates depleted with negative control or ICE1 siRNAs were subjected to co-immunoprecipitation with IgG or CASC3 antibodies. Antibodies against UPF and EJC proteins were used for detection, as indicated on the left. (Right) Quantification of three independent experiments showing densitometry values of UPF3B or eIF4AIII recovery during CASC3 i.p. with or without ICE1 depletion. Error bars represent standard deviation, and the asterisk indicates statistical significance compared to control (Student's t-test, p<0.05). (**B**) (Left) Imaging flow cytometry of 293 cells stably expressing GFP-UPF3B fusion protein during ICE1 or control depletion. Images presented represent unbiased collection events generated by IDEAS software from each treatment group's median bin. (Top right) Quantification of nuclear/cytosolic GFP-UPF3B fusion protein ratios during ICE1 depletion or control. Bars represent 25th, 50th, and 75th quartile for each treatment group, and

*Figure 6 continued on next page*

*Figure 6 continued*

asterisks indicate statistical significance (n = 1588 (siNT); n = 2167 (siICE1); Mann Whitney test, p<10–15). (Bottom right) Western blot of total GFP-UPF3B protein levels during ICE1 depletion.

DOI: https://doi.org/10.7554/eLife.33178.019

The following figure supplements are available for figure 6:

**Figure supplement 1.** Overexpression of the ICE1 MIF4G domain disrupts interaction of UPF3B with eIF4AIII.

DOI: https://doi.org/10.7554/eLife.33178.020

**Figure supplement 2.** An EJC-binding mutant of GFP-UPF3B is enriched in the nucleus.

DOI: https://doi.org/10.7554/eLife.33178.021

localization of the wild-type GFP-UPF3B and a mutant protein partially defective for EJC binding (GFP-UPF3BΔ412-434 [*Chamieh et al., 2008*; *Gehring et al., 2003*]). The EJC-binding impaired UPF3B exhibited a modest defect in co-immunoprecipitation with eIF4AIII and significantly enhanced nuclear localization, consistent with the results from ICE1 knockdown cells (*Figure 6—figure supplement 2*). These findings suggest that inability of UPF3B to interact with EJCs may in part account for lack of export to the cytoplasm in the absence of ICE1.

## UPF3B overexpression overcomes ICE1 depletion

Previous studies have shown that a UPF3B peptide can interact directly with EJCs assembled in vitro, albeit with modest affinity (~10 µM; [*Buchwald et al., 2010*]). Our data suggest that ICE1 promotes association of UPF3B with the EJC, raising the possibility that ICE1 depletion may be overcome by boosting UPF3B levels. To test this hypothesis, we depleted ICE1 from parental HEK-293 cells or a cell line stably over-expressing 3XFLAG-UPF3B and assessed the ability of CASC3 to co-purify tagged and endogenous UPF3B. As shown in (*Figure 6A*), ICE1 depletion from parental cells caused a decrease in UPF3B recovery with CASC3 (*Figure 7A*, lanes 2 and 4). However, this defect in association could be rescued by overexpressing 3XFLAG-UPF3B (*Figure 7A*, lanes 6 and 8). To determine whether restoration of the EJC-UPF3B interaction could also enhance NMD, we assayed the effect of ICE1 depletion on several NMD target mRNAs in parental and UPF3B-overexpression lines. As above, ICE1 depletion caused an increase in levels of ATF3, GADD45B, GAS5, and ANXA1 mRNAs. Importantly, over-expressing exogenous UPF3B partially rescued the NMD phenotype, causing significant reductions in the abundance of the EJC-mediated substrates (*Figure 7B*). Consistent with previous findings, overexpression of UPF3B in the presence of ICE1 did not affect NMD target levels (*Huang et al., 2011*), suggesting that the rescue of ICE1 depletion was not simply due to enhanced NMD efficiency. Together, the biochemical and functional rescue of ICE1 depletion by UPF3B over-expression indicates that the loss of NMD in the absence of ICE1 is due to a failure in UPF3B-EJC assembly or stability.

## Discussion

Beginning from whole-genome RNAi screening for proteins involved in the human NMD pathway, we present evidence that ICE1 is a EJC-associated protein that promotes UPF3B-EJC association and regulation of a large swath of NMD targets. This discovery was enabled by the CSA approach, which minimizes the impact of off-target effects mediated by siRNA seed sequences (*Marine et al., 2012*). Indicative of the potential to identify novel NMD factors, this strategy resulted in high levels of enrichment of proteins known to be involved in NMD. The screen was particularly well suited to identification of factors involved in EJC-enhanced NMD, with all four core EJC components and two previously known assembly factors exhibiting high median seed-corrected Z-scores, in addition to the newly identified EJC-interacting factor ICE1. We focus on the role of ICE1 in this study, but we expect that the screen data, particularly when combined with a recent CRISPR-based screen (*Alexandrov et al., 2017*), will provide a valuable resource for further investigation of novel NMD factors.

ICE1 depletion results in increased abundance of mRNAs selected for NMD on the basis of stop codons that violate the 50–55 nt rule, uORFs, and 3'UTR length. While decay of the former class is clearly stimulated by the presence of EJCs, the uORF class likely comprises a mix of transcripts that undergo EJC-dependent and EJC-independent decay, depending on transcript architecture, uORF

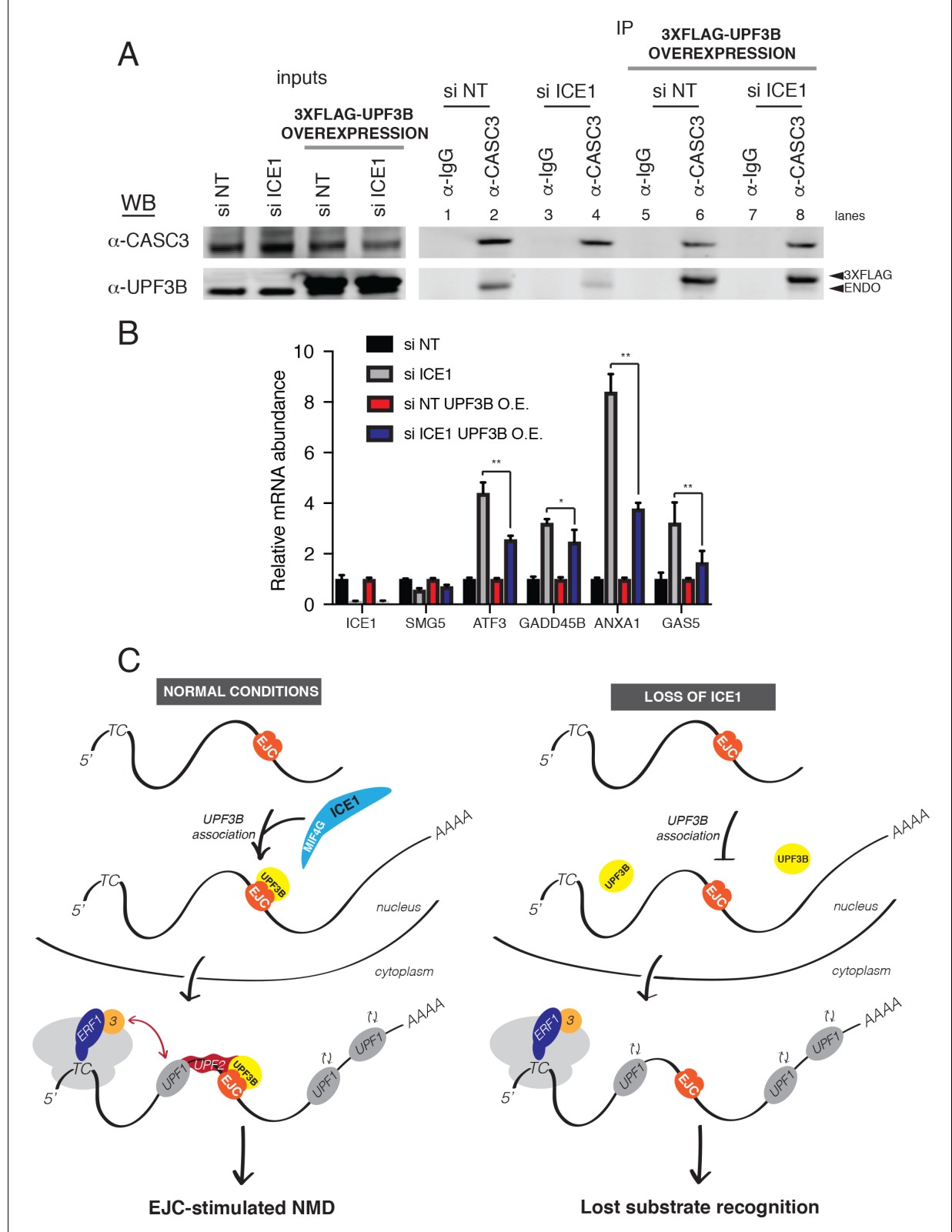

**Figure 7.** UPF3B overexpression ameliorates NMD defect during ICE1 depletion. (**A**) Western blot of cell lysates engineered to stably overexpress 3XFLAG-UPF3B or parental vector treated with ICE1 or non-targeting control siRNAs. Co-immunoprecipitation of the EJC was performed using anti-CASC3 antibodies. Input lanes represent 3% of the total immunoprecipitated material. (**B**) RT-qPCR analysis of EJC-stimulated targets in parental or UPF3B overexpression cell lines depleted of ICE1 or a non-targeting control. Asterisks indicate statistical significance in fold change between parental

*Figure 7 continued on next page*

*Figure 7 continued*

and UPF3B overexpression lines during ICE1 depletion (n = 3, Student's t-test, *p<0.05; **p<0.01). (**C**) Model illustrating the requirement of ICE1 to facilitate UPF3B association with the EJC and subsequently target EJC-stimulated NMD substrates for decay.

DOI: https://doi.org/10.7554/eLife.33178.022

translation frequency, and other factors. In our RNAseq studies, the largest effects of ICE1 loss appear to be on EJC-enhanced NMD, but our data also suggest a role for the protein in promoting decay of transcripts with longer than normal 3'UTRs. Even among the core NMD factors UPF2 and UPF3B, differential substrate preferences have been observed, leading to the proposal that there are multiple 'branches' of the NMD pathway, each with distinct cofactor requirements (*Chan et al., 2007*; *Gehring et al., 2005*; *Huang et al., 2011*; *Metze et al., 2013*). As yet, however, the RNA features underlying susceptibility to the various proposed NMD sub-pathways are not understood.

Biochemical and functional assays suggest that ICE1 functions in NMD by promoting the proper association of UPF3B with EJCs (*Figure 7C*). UPF2 association with EJCs is also decreased in ICE1 knockdown cells, presumably due to the inability of UPF3B to act as a bridging factor (*Chamieh et al., 2008*). In addition to the defect in UPF3B-EJC association, we also observed reduced accumulation of UPF3B in the cytoplasm of cells depleted of ICE1 (*Figure 6B*). Since UPF3B assembly with EJCs is thought to occur in the nucleus, enhanced UPF3B nuclear localization is unlikely to account for the observed defect in EJC binding. Instead, our data indicate that reduced export of UPF3B may be in part a consequence of decreased association with EJCs in the nucleus, preventing export of UPF3B with mRNPs. Alternatively, decreased stability of the UPF3B-EJC association could result in more rapid re-import of UPF3B. In turn, the reduced abundance of UPF3B in the cytoplasm may have the side-effect of impairing long 3'UTR-mediated NMD, explaining the apparent protection of such targets upon ICE1 depletion. Promotion of efficient UPF3B nuclear export by the EJC could also be necessary for its newly identified function in translation termination and help to explain why depletion of EJC factors has been observed to affect the decay of well-characterized 3'UTR length-dependent NMD targets (*Huang et al., 2011*; *Neu-Yilik et al., 2017*).

Together, our data are consistent with a model in which ICE1 helps to prepare newly assembled EJCs for association with UPF3B (*Figure 7C*). We find that ICE1 interacts with EJC components at endogenous levels, but have not observed an interaction between the endogenous ICE1 and UPF3B proteins. As these experiments were carried out under native conditions, we cannot exclude the possibility that this is in part due to dissociation during isolation. Arguing against this scenario, we readily observe UPF3B co-immunoprecipitation with the EJC under the same purification conditions and can detect co-purification of UPF3B with ICE1 when either UPF3B or the ICE1 C-terminus is overexpressed. Overexpressed UPF3B lacking the ability to bind EJCs also fails to associate with ICE1, leading us to propose that ICE1 and UPF3B can concurrently interact with the EJC but that this complex is scarce or absent under normal cellular conditions. Together, our data suggest that ICE1 transiently interacts with fully assembled EJCs to promote UPF3B-EJC association. Possible effects of ICE1 binding to EJCs could be increased efficiency of EJC maturation or altered post-translational modification of EJC proteins, each of which could have the effect of increasing UPF3B recruitment or the stability of the complex. Alternatively, transient interactions between ICE1 and UPF3B could enhance UPF3B recruitment to EJCs.

With increasing organismal complexity, the NMD pathway has evolved to use the exon junction complex to more efficiently discriminate between aberrant and normal mRNAs. The addition of components to the pathway in turn presents new opportunities for regulatory fine-tuning of decay. Interestingly, duplication of the ancestral UPF3 gene to yield UPF3A and UPF3B proteins has recently been shown to enable cell-type-specific control of NMD (*Shum et al., 2016*). UPF3A has a reduced capacity to interact with EJCs but retains important UPF2-interacting residues, allowing it to disrupt UPF3B activity by competing for UPF2 binding. Our observations suggest that the association of UPF3B with the EJC is also a potential target for regulatory control of NMD. In this case, interference with ICE1 function appears to leave the UPF2-UPF3B interaction intact (*Figure 6—figure supplement 1*), while reducing the ability of UPF3B to associate with EJCs (*Figure 6* and *Figure 6—figure supplement 1*). Since UPF3A and ICE1 affect distinct UPF3B interactions, they could be independently or concurrently used for cell-type- or condition-specific regulation of NMD. Further work will be required to

understand whether ICE1 is subject to regulation, but our findings clearly point to an opportunity for cells and/or therapeutic interventions to manipulate ICE1 to decouple the EJC's role in NMD from its contributions to pre-mRNA splicing, export, localization, and translation.

# Materials and methods

**Key resources table**

| Reagent type (species) or resource | Designation | Source or reference | Identifiers | Additional information |
|---|---|---|---|---|
| Gene (*Homo sapiens*) | ICE1 | Addgene plasmid # 49428 | NM_015325 | Originally KIAA0947 |
| Cell line (*H. sapiens*) | pKC-4.06 | This study | | HEK-293 cell line stably expressing pCMV-3XFLAG-Fluc-B-globin (39PTC) |
| Cell line (*H. sapiens*) | Flp-In T-Rex-293 cells | Invitrogen | Cat. No. R78007 | |
| Cell line (*Drosophila melanogaster*) | S2 cells | ThermoFisher Scientific | Cat. No. R690-07 | |
| Transfected construct (*H. sapiens*) | pCMV-3XFLAG-Fluc-B-globin (39PTC) | Professor Lynne Maquat | NA | |
| Transfected construct (*H. sapiens*) | pcDNA5-FRT-TO-3XF-ICE1 | This study | NA | Cloned using Addgene plasmid # 49428 |
| Transfected construct (*H. sapiens*) | pcDNA5-FRT-TO-3XF-ICE1-Nterm | This study | NA | |
| Transfected construct (*H. sapiens*) | pcDNA5-FRT-TO-3XF-ICE1-MIF4G | This study | NA | |
| Transfected construct (*H. sapiens*) | pcDNA5-FRT-TO-3XF-UPF3B | This study | NA | Cloned from HEK-293 cDNA library |
| Transfected construct (*H. sapiens*) | pcDNA5-FRT-TO-3XMYC-UPF3B | This study | NA | |
| Transfected construct (*H. sapiens*) | pcDNA5-FRT-TO-3XMYC-UPF3B(delN-term) | This study | NA | |
| Transfected construct (*H. sapiens*) | pcDNA5-FRT-TO-3XMYC-UPF3B(delC-term) | This study | NA | |
| Transfected construct (*H. sapiens*) | pcDNA5-FRT-TO-3XMYC-UPF3B(del421-434) | This study | NA | |
| Transfected construct (*H. sapiens*) | pcDNA5-FRT-TO-GFP-UPF3B | This study | NA | |
| Transfected construct (*H. sapiens*) | pcDNA5-FRT-TO-GFP-UPF3B(del421-434) | This study | NA | |
| Antibody | anti-ACTIN | Cell Signaling | Cat. No. 3700S, RRID:AB_2242334 | (1:5000) |
| Antibody | anti-ANXA1 | Bethyl Laboratories | Cat. No. A305-235A, RRID:AB_2631628 | (1:1000) |
| Antibody | anti-ATF3 | Abcam | Cat. No. Ab58668, RRID:AB_879578 | (1:500) |
| Antibody | anti-CASC3 | Bethyl Laboratories | Cat. No. A302-472A, RRID:AB_1944210 | (1:1000) |
| Antibody | anti-CASC3 | Bethyl Laboratories | Cat. No. A302-471A, RRID:AB_1944212 | For IP |
| Antibody | anti-eIF4AIII | Bethyl Laboratories | Cat. No. A302-981A, RRID:AB_10748369 | (1:1000) |
| Antibody | anti-eIF4AIII (mouse monoclonal) | Developmental Studies Hybridoma Bank | PCRP-EIF4A3-3A2 | For IP |
| Antibody | anti-FLAG | Sigma | Cat. No. F1804, RRID:AB_262044 | (1:1000) |

*Continued on next page*

*Continued*

| Reagent type (species) or resource | Designation | Source or reference | Identifiers | Additional information |
|---|---|---|---|---|
| Antibody | anti-GADD34 | Proteintech | Cat. No. 10449–1-AP, RRID:AB_2168724 | (1:500) |
| Antibody | anti-GADD45B | Abcam | Cat. No. Ab105060, RRID:AB_10714129 | (1:500) |
| Antibody | anti-KIAA0947 (ICE1) | Bethyl Laboratories | Cat. No. A304-276A, RRID:AB_2620472 | (1:1000) |

## RNAi screen

To generate an NMD-sensitive luciferase reporter, we constructed pCMV-3XFLAG-FLuc-β-globin (39PTC) using PCR, traditional cloning methods, and β-globin reporters kindly provided by Professor Lynne Maquat (University of Rochester). Pilot screens were performed by transfecting HEK-293 cells with pCMV-3XFLAG-FLuc-β-globin (39PTC) using Lipofectamine 2000 according to the manufacturer's protocol (Thermo Fisher Scientific, Waltham, MA). 24 hr after transfection, selection for stable genomic integration was performed with 800 μg/mL Geneticin (Thermo Fisher Scientific) in 96-well plates at three cells/well density. Two weeks following the geneticin selection, 33 monoclonal cell colonies were isolated and expanded. The colony that showed the best response to the UPF1 siRNAs (denoted pKC-4.06) was selected and expanded for use in the global assay. Whole-genome RNAi screening was conducted using the facilities of the Division of Pre-Clinical Innovation at NCATS (Rockville, MD). Considerations for design, optimization, analysis and hit selection criteria of the RNAi assay were taken according to the extensive guidelines previously outlined (*Auld and Inglese, 2016*; *Martin et al., 2012*). Briefly, siRNA screening was conducted using a genome-wide library of Silencer Select siRNAs (Thermo Fisher Scientific) comprising three siRNAs per gene for ~21,000 genes. 2 pmol of siRNA (20 nM final concentration) was pre-spotted to 384-well plates, and 0.07 μL Lipofectamine RNAiMax was added in 20 μL of serum-free DMEM media. This complex was incubated at ambient temperature for 30 min prior to adding 1000 pKC-4.06 cells in 20 μL of 20% serum DMEM media. Cells were incubated for 72 hr prior to the addition of OneGlo (Promega, Madison, WI) luciferase assay reagent. For data analysis, screen data were filtered for off-target effects by applying the Common Seed Analysis (CSA) approach (*Marine et al., 2012*). Hit selection was performed by converting normalized values into ranked Z-scores, and statistical significance determined with non-parametric Wilcoxon rank sum tests. STRING was used to identify any known interactions among the top screen hits (median seed corrected Z > 1.5; [*Szklarczyk et al., 2015*]). For comparison to the siRNA screen, high-throughput sequencing data (NCBI SRA BioProject accession PRJNA353310) from a previously reported pooled CRISPR screen for NMD components were analyzed with MAGeCK software (*Alexandrov et al., 2017*; *Li et al., 2014*).

## Cell culture and endogenous gene depletion by RNAi

Human Flp-In T-REx-293 cells (Invitrogen, Carlsbad, CA; Cat. No. R78007) were maintained at 37°C and 5% $CO_2$ in DMEM with 10% FBS and 1% pen/strep. Cells were obtained directly from the manufacturer and periodically assayed for mycoplasma contamination. Gene depletion studies were carried out by reverse transfection with siRNA non-targeting control (siNT; Thermo Fisher Scientific, Silencer Select Negative Control #2), siUPF1 (5'- AAGATGCAGTTCCGCTCCATTTT-3'; (*Mendell et al., 2004*), siICE1 (5'-GGAAGATGATTATTCGTTATT-3'), and siUPF3B (5'-GGAGAAGC-GAGTAACCCTG-3'; (*Kim et al., 2005*) as previously described (*Ge et al., 2016*). Briefly, 25 pmols of gene-specific or non-targeting siRNA was directly pipetted into each well of a Falcon six-well flat bottom multiwell cell culture plate (Corning, Corning, NY; Cat. No. 353046). Reverse transfections were then conducted using Lipofectamine RNAiMAX reagent according to the manufacturer's protocol (Thermo Fisher Scientific, Cat. No. 13778150). As the transfection master mix is comprised of Opti-MEM reduced serum medium (Thermo Fisher Scientific Cat. No. 31985062), an equal volume of DMEM with 20% FBS with pen/strep was used to plate cells and attain a final 10% FBS concentration. 72 hr following the transfection, lysates were collected for protein or RNA isolation.

## RNA isolation, cDNA preparation, and RT-qPCR

Total RNA was extracted and purified from whole-cell lysates using the RNeasy Mini Kit with on-column DNase digestion (Qiagen, Hilden, Germany; Cat. No. 74106). RNA concentration was determined on a NanoDrop 1000 (Thermo Fisher Scientific), and 1 µg of RNA used as a template for cDNA library preparation using the Maxima First Strand cDNA synthesis Kit for RT-qPCR (Thermo Fisher Scientific, Cat. No. K1641). The resulting cDNA libraries were further diluted 20-fold with ultrapure H$_2$O and subsequently analyzed by RT-qPCR using iTaq Universal SYBR Green Supermix (BioRad Laboratories, Hercules, CA;, Cat. No. 1725124) on a Roche LightCycler 96 instrument (Roche Diagnostics Corporation, Indianapolis, IN). Sequences for gene-specific primers used for amplification are listed in *Supplementary file 7*. 2$^{-\Delta\Delta CT}$ values were calculated using GAPDH or b-ACTIN for normalization, and all reported values are representative of three independent biological replicates (*Ge et al., 2016*).

## Plasmids

pcDNA5-Flag-KIAA0947 (ICE1) was a gift from Joan Conaway and Ronald Conaway (Addgene plasmid # 49428; [*Takahashi et al., 2011*]). Phusion High-Fidelity DNA Polymerase (New England Biolabs, Cat. No. M0530) was used to PCR amplify ICE1 and UPF3B sequences from plasmid # 49428 and human cDNA, respectively, and cloned into pcDNA5-FRT-TO-3XF using traditional methods. Deletion constructs were generated using diverging 5'-phosphorylated primers followed by ligation with T4 DNA Ligase (New England Biolabs, Cat. No. M0202S), and all constructs were sequence validated.

## Metabolic labeling and mRNA half-life quantification

To determine specific mRNA half-life measurements during ICE1 and UPF1 depletion, HEK-293 cells were reverse transfected with a gene-specific or non-targeting control siRNA for 72 hr. At the end of the depletion, cells were treated with 0.2 mM 5-ethynyl Uridine (5-EU) for 60 mins. Cells were then immediately harvested and RNA was isolated and column purified using the RNeasy Mini Kit with on-column DNase digestion (Qiagen, Hilden, Germany; Cat. No. 74106). In addition, *Drosophila* S2 cells (ThermoFisher Scientific, Cat. No. R690-07) were cultured at ambient temperature for 24 hr in Sf-900 media (ThermoFisher Scientific, Cat. No. 10967032) containing 0.1 mM 5-EU, and RNA was isolated and processed as with the human cells for use as a spike-in control. The *Drosophila* spike in control was then quantified and later used to account for variations in biotinylation efficiency and recovery on the streptavidin beads. To differentiate total and nascent RNA levels in each sample, 2 µg of sample RNA was combined with 200 ng (10%) of the spike-in control and partitioned using the Click-iT Nascent RNA Capture Kit (ThermoFisher Scientific, Cat. No. C10365) following the manufacturer's protocol. 1 µg of biotinylated RNA from each sample was used for 'total' cDNA library preparation, with the remaining 1 µg of RNA from that sample applied to the Streptavidin T1 magnetic beads for labeled RNA capture and cDNA synthesis. The resulting 50 µL cDNA libraries were diluted, and mRNA abundance was determined using qRT-PCR as previously described. Nascent RNA recovery was normalized to *Drosophila* RP49 mRNA levels, and individual half-lives were determined using the equation: $t_{1/2}$ = -$t_L$ * $ln(2)/ln(1/R)$, where $t_L$ is the EU labeling time in minutes and $R$ is the abundance in nascent RNA fraction/abundance in total RNA fraction (*Dölken, 2013*; *Haque et al., 2018*; *Russo et al., 2017*).

## RNAseq

Total RNA from HEK-293 cells transfected with siICE1, siUPF1, siUPF3B, or siNT as above was assessed using an Agilent Bioanalyzer 2100, subjected to ribosomal RNA removal using the Ribo-Zero rRNA Removal Kit (Illumina, San Diego, CA), and used for library preparation with the Illumina TruSeq Stranded Total RNA Sample Preparation Kit (Illumina). Paired-end 50 nt reads were generated on the Illumina HiSeq 2000 platform, and the resulting reads were mapped using HISAT2 software to the GRCh37/hg19 combined genome and transcriptome indexes provided by the authors (*Kim et al., 2015*). Mapped reads and Ensembl gene models were used for transcriptome assembly by StringTie (*Pertea et al., 2015*), after which individual assemblies were merged using TACO (*Niknafs et al., 2017*) to obtain a higher-confidence annotation better reflecting HEK-293 expression patterns. The resulting TACO-curated transcriptome was used for pseudoalignment-based transcript

quantification and analysis of differential gene and isoform usage with Kallisto/Sleuth (*Bray et al., 2016*; *Pimentel et al., 2017*). To identify transcript isoforms violating the 50–55 nt rule, the longest ORF in each TACO-derived transcript was annotated with IsoformSwitchAnalyzeR (*Vitting-Seerup and Sandelin, 2017*). The most highly expressed isoform from each gene, as judged by the transcripts per million (TPM) Kallisto calculation, was used to compute 3'UTR lengths for down-stream gene-level analyses.

Changes in RNA stability were inferred from RNAseq data using REMBRANDTS (*Alkallas et al., 2017*), following HISAT2 mapping to the GRCh38 combined genome and transcriptome indexes provided by the authors (*Kim et al., 2015*). Reads mapping to constitutive exons and introns of Ensembl GRCh37.87 annotations were quantified with HTSeq (*Anders et al., 2015*), and a read cut-off stringency of 0.99 was used for REMBRANDTS analysis. Majiq and Leafcutter were used to analyze splicing changes following siICE1 and siNT treatment (*Li et al., 2018*; *Vaquero-Garcia et al., 2016*). For Majiq, a change in isoform usage of 10% or greater at the 95% confidence level was used to identify genes potentially undergoing alternative splicing. For Leafcutter, genes with a change in isoform usage of 10% or greater and adjusted $p < 0.05$ were selected. Raw and processed RNAseq data are available in the NCBI GEO database, accession GSE105436.

## Cell extract preparation, immunopurifications and western blotting

A freeze/thaw lysis protocol was implemented to harvest lysates for protein assays, as described (*Hogg and Collins, 2007*). Briefly, cells were collected in 15-mL conical tubes following trypsinization and resuspension in media, centrifuged at 500 xg for 5 min at 4°C, and rinsed once with 1 mL ice-cold PBS. Lysates were then transferred to a 1.5 mL Eppendorf tube and centrifuged at 2000 xg for 1 min at 4°C. Cell pellet volume was estimated with an analytical balance, and the pellet resus-pended in 5X packed cell volume of ice-cold hypotonic lysis buffer (e.g., 500 µL of lysis buffer for a 100 µL cell pellet; buffer contains 20 mM HEPES, pH 7.4, 2 mM $MgCl_2$, 10% glycerol, 1 mM DTT, 1X Protease and Phosphatase Inhibitor Cocktail, Thermo-Fisher Cat. No. 78440). Following a 5 min incu-bation on ice, cell extracts were snap frozen in liquid nitrogen, briefly thawed in a 37°C $H_2O$ bath, and frozen and thawed again. With the second and final thaw, 5M NaCl was added to achieve a final 150 mM concentration, and extracts were allowed to incubate on ice for 5 min. To isolate the soluble fraction, extracts were centrifuged for 15 min at 20,000 x g at 4°C, and the supernatant was ali-quoted to fresh 1.5 mL Eppendorf tubes for a final snap freeze in liquid nitrogen prior to storage at −80°C.

Western blots were performed using the NuPAGE electrophoresis and transfer systems (Invitro-gen). Denatured proteins were resolved on 3–8% Tris-acetate or 4–12% bis-Tris gels depending on molecular weight, and transferred to nitrocellulose membranes according to the NuPAGE manufac-turer's protocol (Invitrogen). Membranes were incubated on an orbital shaker with blocking buffer for fluorescent western blotting (Rockland Immunochemicals, Limerick, PA) for 1 hr at room temper-ature. Incubations with the primary antibody were performed overnight at 4°C on an orbital shaker. Following three 10 min washes with 1XTBS (0.1% Tween-20), membranes were incubated with the appropriate secondary antibody for one hour at room temperature, followed by three additional 10 min washes with 1XTBS. Final quantitative western blot images were obtained on a Licor Odyssey imaging system (LI-COR Biosciences, Lincoln, NE). All antibodies used for immunoprecipitation and/or western blotting are provided in the Key Resources table. The eIF4AIII hybridoma cell line was developed by the Protein Capture Reagents Program and obtained from the Developmental Studies Hybridoma Bank (DHSB), created by the NICHD of the NIH and maintained at The University of Iowa, Department of Biology, Iowa City, IA 52242.

## Isolation and purification of monoclonal eIF4AIII antibody

A murine hybridoma cell line (PCRP-EIF4A3-3A2-f) that expresses secreted monoclonal antibody against the full-length human eIF4AIII epitope was obtained from the Developmental Studies Hybrid-oma Bank at the University of Iowa. Hybridoma cultures were maintained at 37°C and 5% $CO_2$ in DMEM (Gibco) supplemented with 1% pen/strep and 10% Ultra Low IgG FBS (ThermoFisher Scien-tific, Catalog # 16250078). Cultures were visually inspected for overall health and divided while the cells were still in log phase growth, or approximately $5 \times 10^5$ - $1 \times 10^6$ cells/mL. To purify monoclo-nal eIF4AIII antibody from the culture supernatant, 50 mL of culture media from cells in log phase

growth was centrifuged at 4°C for 5 min at 1000 RPM. The supernatant was then sterile filtered using a 0.4 mm filter and stored at 4°C overnight. Prior to affinity purification, a protein G column (HiTrap Protein G HP 1 mL, Catalog # 339-0485-81) was equilibrated with 10 column volumes of 100 mM Tris-HCl pH 8.0. Meanwhile, the pH of the cell culture supernatant was adjusted by adding 5 mL (~10% total volume) of 1.0M Tris-HCl. The total supernatant volume was passed through the protein G column attached to an AKTApurifier 10 (G.E.) at a speed of 1 mL/min. After loading the sample, the column was washed with 10 column volumes of 100 mM Tris-HCl, followed by another 10 column volumes of 10 mM Tris-HCl. The antibody was then eluted from the column with 50 mM glycine pH 3.0, and collected into Falcon tubes containing a neutralization buffer of 100 mL 1M Tris-HCl. Fractions containing the antibody were determined by monitoring $UV_{280}$ values during the elution process, and purity and concentration determined by Coomassie staining and Pierce 660 calorimetric protein assay reagent (Catalog # 22660), respectively.

## Imaging flow cytometry

eGFP-UPF3B and eGFP-UPF3B$\Delta$412–434 expression constructs were cloned into a tetracycline-inducible pcDNA5 vector and integrated into human Flp-In T-REx-293 cells. Gene-specific depletions for ICE1 and a non-targeting control siRNA were performed as described above. At the time of transfection, eGFP-fusion protein expression was induced by the addition of doxycycline to a final concentration of 200 ng/mL. Prior to harvesting the samples, Hoechst dye was added directly to the media to a final concentration of 10 µg/mL for 45 min at 37°C. At the end of the incubation period, cells were trypsinized, rinsed with 1X PBS, and resuspended in 50 µL of 1X PBS with 2% FBS in low protein binding tubes. Spectral compensation controls including non-fluorescent parentals, GFP-only, and Hoechst-only labeled cells were processed in parallel to allow for subsequent data acquisition and analysis. Data were collected and analyzed on an ImageStream imaging flow cytometer (Amnis, Millipore Sigma, Seattle, WA), according to the manufacturer's protocol. Briefly, data were collected using INSPIRE software (Amnis) and 405 nm and 488 nm lasers were used to excite Hoechst and eGFP, respectively. Laser powers were chosen in order to prevent pixel saturation, and 10,000 single and focused events were captured per experimental condition. Single color compensation controls were merged to generate a compensation matrix and all sample files were analyzed with this matrix applied. Data analysis was performed with IDEAS software (version 6.2, Amnis), with eGFP expression levels or 'intensity' calculated by the software as the sum of pixel values minus the background pixel values. To determine nuclear localization of the eGFP signal, a morphology mask was created to conform to the shape of the nuclear Hoechst imagery. For determining the cytoplasmic area, a combined mask that subtracted the nuclear Hoechst imagery from an erode mask of the brightfield image of the total cell area was made. The intensity of GFP was then calculated by the ratio of GFP expression in the nucleus to the cytoplasmic region.

## Acknowledgements

We thank Prof. Lynne Maquat of the University of Rochester for kindly providing the β-globin plasmids used to construct the NMD luciferase fusion reporter. We also thank former and current members of the Hogg lab and Lisa Postow for helpful discussions and critical reading of the manuscript. In addition, we are grateful to Tania Nguyen and Stacey Baker for piloting gene depletion studies, Nazmul Haque for performing siRNA transfection and RNA isolation for RNAseq, and Todd Schoborg for helpful suggestions on *Drosophila* S2 cell culture. High-throughput sequencing was conducted by Jun Zhu and Poching Liu in the NHLBI DNA Sequencing Core. Venina Dominical and the NHLBI Flow Cytometry Core assisted with the imaging flow cytometry experiments. This work was supported by the NIH Intramural Research Programs of NHLBI and NCATS and used the computational resources of the NIH HPC Biowulf cluster.

## Additional information

### Funding

| Funder | Grant reference number | Author |
| --- | --- | --- |
| National Heart, Lung, and Blood Institute | Intramural Research Program | Thomas D Baird<br>J Robert Hogg |
| National Center for Advancing Translational Sciences | Intramural Research Program | Ken Chih-Chien Cheng<br>Yu-Chi Chen<br>Eugen Buehler<br>Scott E Martin<br>James Inglese |

The funders had no role in study design, data collection and interpretation, or the decision to submit the work for publication.

### Author contributions

Thomas D Baird, Conceptualization, Data curation, Formal analysis, Investigation, Visualization, Methodology, Writing—original draft, Writing—review and editing, Designed, performed, and analyzed all experiments following the siRNA screen, Wrote the manuscript; Ken Chih-Chien Cheng, Conceptualization, Data curation, Formal analysis, Investigation, Methodology, Writing—review and editing, Conceived of and designed the assay used for high-throughput screening, Constructed and characterized the NMD reporter cell line, Assisted with preparation of the manuscript; Yu-Chi Chen, Investigation, Methodology, Helped to conduct the RNAi screen; Eugen Buehler, Conceptualization, Data curation, Software, Formal analysis, Validation, Investigation, Methodology, Designed and analyzed the RNAi screen; Scott E Martin, Conceptualization, Data curation, Software, Formal analysis, Supervision, Validation, Investigation, Visualization, Methodology, Writing—review and editing, Designed and analyzed the RNAi screen, Assisted with preparation of the manuscript; James Inglese, Conceptualization, Resources, Data curation, Formal analysis, Supervision, Funding acquisition, Visualization, Methodology, Project administration, Writing—review and editing, Conceived of and designed the assay used for high-throughput screening, Designed and analyzed the RNAi screen, Assisted with preparation of the manuscript; J Robert Hogg, Conceptualization, Resources, Data curation, Software, Formal analysis, Supervision, Funding acquisition, Visualization, Methodology, Writing—original draft, Project administration, Writing—review and editing, Designed gene depletion and high-throughput sequencing experiments and analyzed RNAseq data, Designed and analyzed all experiments following the siRNA screen, Wrote the manuscript

### Author ORCIDs

Thomas D Baird (iD) https://orcid.org/0000-0002-0280-9984
James Inglese (iD) http://orcid.org/0000-0002-7332-5717
J Robert Hogg (iD) http://orcid.org/0000-0001-5729-5135

### Decision letter and Author response

Decision letter https://doi.org/10.7554/eLife.33178.036
Author response https://doi.org/10.7554/eLife.33178.037

## Additional files

### Supplementary files

• Supplementary file 1. Results of whole-genome siRNA screen (xlsx).
DOI: https://doi.org/10.7554/eLife.33178.023

• Supplementary file 2. Spliceosomal proteins among top hits in siRNA screen (xlsx).
DOI: https://doi.org/10.7554/eLife.33178.024

• Supplementary file 3. Results of secondary siRNA screen of selected genes from primary screen (xlsx).
DOI: https://doi.org/10.7554/eLife.33178.025

• Supplementary file 4. Differential gene expression analysis of HEK-293 cells treated with siICE1, siUPF1, and siNT (xlsx).
DOI: https://doi.org/10.7554/eLife.33178.026

• Supplementary file 5. Differential splicing analysis of HEK-293 cells treated with siICE1 and siNT (xlsx).
DOI: https://doi.org/10.7554/eLife.33178.027

• Supplementary file 6. Relative mRNA stability analysis of HEK-293 cells treated with siICE1, siUPF1, siUPF3B, and siNT (xlsx).
DOI: https://doi.org/10.7554/eLife.33178.028

• Supplementary file 7. RT-qPCR primer sequences (xlsx)
DOI: https://doi.org/10.7554/eLife.33178.029

• Transparent reporting form
DOI: https://doi.org/10.7554/eLife.33178.030

### Major datasets

The following datasets were generated:

| Author(s) | Year | Dataset title | Dataset URL | Database, license, and accessibility information |
|---|---|---|---|---|
| Baird TD, Cheng KC-C, Chen Y-C, Buehler E, Martin SE, Inglese J, Hogg JR | 2018 | ICE1 promotes the link between splicing and nonsense- mediated mRNA decay | https://www.ncbi.nlm.nih.gov/geo/query/acc.cgi?acc=GSE105436 | Publicly available at the NCBI Gene Expression Omnibus (accession no: GSE105436) |
| Alexandrov A, Shu MD, Steitz JA | 2017 | Screening GeCKO lentiCRISPR knockout sgRNA library in human Fireworks HeLa cells to identify components and regulators of the human nonsense-mediated mRNA decay pathway | https://www.ncbi.nlm.nih.gov/bioproject/PRJNA353310/ | Publicly available at NCBI BioProject (accession no: PRJNA353310) |

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
