## [Decision Letter]

Thank you for submitting your article "ICE1 promotes the link between splicing and nonsense-mediated mRNA decay" for consideration by *eLife*. Your article has been favorably evaluated by James Manley (Senior Editor) and three reviewers, one of whom is a member of our Board of Reviewing Editors. The reviewers have opted to remain anonymous.

The reviewers have discussed the reviews with one another and the Reviewing Editor has drafted this decision to help you prepare a revised submission.

We have received comments from three experts in the field all of whom found the manuscript to present new and important information about the protein ICE1 revealed in the high throughput screen for factors critical for degradation of NMD substrates. Despite broad enthusiasm, the reviewers agree that several issues need to be addressed experimentally, in addition to changes in the text to rephrase the conclusions drawn from the experiments more cautiously. In case of modest effects, softer wordings such as "the data indicate/are consistent with/support the view" seem more appropriate than "demonstrate". Most critically, while it is clear that ICE1 impacts the amount of NMD-target mRNAs in the cell disproportionately relative to non-NMD-target mRNAs, the mechanism for this effect is not wholly clear. The reviewers recommend that the authors establish that ICE1 depletion does not alter splicing or block export from the nucleus, thus stabilizing the mRNA, and that mRNA stability is impacted and not simply overall levels (i.e. a decay experiment needs to be performed).

All reviewers had issues with Figure 4A (see detailed reviews). Finally, the reviewers had concerns about the magnitude and correlation of effects on mRNAs in UPF1 and ICE1 depletes. It seems likely that if the proposed mechanism is true (that ICE1 is involved in deposition of the EJC complex) then the RNA-seq data of the ICE1-delete should be even better correlated with that of the UPF3B-deplete than with the UPF1-deplete.

We hope these comments are helpful in revising your manuscript.

*Reviewer #1:*

In the current work, Baird et al. present the results of a shRNA-based genetic screen for genes involved in nonsense-mediated mRNA decay (NMD) of a luciferase reporter. While a number of their top hits are the same as those found in a previous CRISPR based screen using a fluorescent reporter, most of the hits are non-overlapping. Among the novel candidate NMD factors that they identify is ICE1, which they go on to characterize further. Depletion of ICE1 clearly leads to stabilization of NMD targets making it an exciting novel player in NMD. Further experiments suggest that ICE1 may act by aiding recruitment of the NMD factor UPF3B to the exon junction complex. While the authors make some headway in determining ICE1's mechanistic role in NMD (interactions with UPF3B in part through the MIF4G domain), it is still not clear that ICE's role in the process is direct. This work is an important advance, but some of the experiments require additional controls, and the evaluation of their mechanistic claims will likely hinge on these additional experiments.

1) Where are the UPF proteins in Figure 1C?

2) I found Figure 4A nearly impossible to interpret. What are the bands in the anti-flag panel blot in the anti-flag IP? Non-specific bands should be indicated as well as specific ones in both sides of the panel. It looks like the mock and 3XF-MIF4GICE1 lanes are identical, meaning that there's no evidence that the pulldown worked (or else this is a non-specific band which is lower in the 3XF-UPF3B lane). This is the only evidence for sufficiency of the MIF4G domain for interaction with eIF4AIII, so this figure needs to be clarified – and if these are the main data, the interaction is quite weak. Is MATRIN3 a control for non-specific association?

3) In Figure 5B, the total amount of GFP-UPF3B is not presented, only the nuclear/cytoplasmic ratio. Are the overall levels of UPF3B affected?

4) For the scatterplot in Figure 3—figure supplement 1A, both axes are normalized to the same siNT control dataset, which makes them poor evidence for similarity of UPF1 and ICE1 effects. For 3 arrays generated from a normal distribution (A, B, C), log(A/C) and log(B/C) will be highly correlated most of the time.

5) The last paragraph of the subsection “ICE1 depletion increases abundance of transcripts with NMD-inducing features” overinterprets the data in Figure 2E and Figure 2—figure supplement 1C. I think the simplest explanation for the differences in siICE and siUPF1 would be differences in knockdown efficiency, or perhaps differences in concentration requirements for activity. I don't think that these data represent evidence for a UTR-length dependent activity difference.

6) In Figure 4—figure supplement 1, it seems that there is some residual IP of UPF3B with ICE when UPF3B lacks the EJC-binding domain. Do these mutants display residual EJC binding? Some additional controls, perhaps with EJC factors or interacting proteins are required.

7) The authors argue that the activity of ICE1 is entirely through recruitment of UP3B to the EJC. The ideal experiment to test this would be an epistasis experiment, which may not be possible given the essentiality of the NMD machinery. However, it might be a start to check by RNA-seq that the genome-wide effects of UPF3B knockdown on NMD are similar to the effects of ICE1 on NMD. As it is, all of the genome-wide analysis presented is with comparisons of ICE to UPF1 knockdown.

8) In the Discussion: "interference with ICE1 function leaves the UPF2-UPF3B interaction intact". It is not clear to me from Figure 5A that this statement is true. The UPF3B/UPF2 interaction was not tested upon ICE1 depletion (or not presented). It seems like UPF2 may be more depleted at the EJC than UPF3b upon ICE1 depletion, but this was not quantified.

*Reviewer #2:*

Nonsense-mediated mRNA decay (NMD) detects and promotes the degradation of transcripts containing premature termination codons and other NMD-inducing features (i.e. long 3' UTRs, uORF translation). In metazoans, the efficiency of NMD is significantly enhanced by the presence of an exon junction complex (EJC) downstream of the terminating ribosome. The association of NMD protein UPF3 with EJCs is thought to underlie enhanced NMD by promoting recruitment of UPF2, which together stimulate UPF1 activity on the upstream terminating ribosome.

In this work, the authors perform a genome-wide RNAi screen in HEK-293 cells to identify novel factors required for degradation of NMD substrates, and identify ICE1, a protein previously characterized in promoting snRNA transcription. Notably, RNA-Seq analysis of cells depleted for ICE1 revealed a small increase in abundance of PTC-containing RNAs (over normal mRNA) and a modest elevation of transcripts harboring uORFs and particularly long 3' UTRs. Co-immunoprecipitation experiments demonstrate that ICE1 can interact with EJC core components in an RNA-independent manner, and that this interaction is mediated through a putative MIF4G domain within the C-terminus of the protein.

The authors propose that ICE1 depletion impairs the nuclear assembly of UPF3B into EJCs, resulting in impaired EJC-enhanced degradation of NMD substrates in the cytoplasm. Three pieces of evidence support this model. First, depletion of ICE1 results in the reduced association between UPF3B and EJC-core protein CASC3 by CoIP. Second, in ICE1 knock-down cells, there is an increased accumulation of UPF3B in the nucleus. Third, NMD activity in ICE1-depleted cells is partially rescued by overexpression of UPF3B.

This work proposes a novel function for ICE1 in promoting UPF3 assembly into EJCs and the downstream EJC-enhanced degradation of NMD substrates. While the model is consistent with experimental evidence, is not robustly supported by the data or directly tested. Moreover, additional interpretations could account for the experimental observations.

1) There are a number of pieces of data to suggest that ICE1 may have an independent or additional function in mRNA metabolism outside of promoting UPF3B association with EJCs.

a) While depletion of ICE1 appears to cause a reproducible reduction (but not elimination) in UPF3B association with core EJC components, its depletion increases the abundance of two characterized NMD substrates 4-fold greater than depletion of the core NMD factor, UPF1 (Figure 4B). This is completely unexpected if ICE1 function on these transcripts is through NMD.

b) While over-expression of UPF3B completely restores its ability to associate with EJCs (as measured by CASC3 CoIP; Figure 6A), the abundance of several NMD-sensitive mRNAs is only partially restored (~50%; Figure 6B) – indicating that abrogation of NMD in ICE1-depleted cells is not caused entirely by impaired interaction between UPF3B and the EJC.

c) A function for ICE1 in EJC assembly/remodeling is unexpected given its absence from past biochemical characterizations of EJC components.

2) It is never tested whether ICE1 function on NMD-substrate abundance is directly mediated through the NMD pathway. For example, ICE1 depletion should not alter mRNA levels in cells also inhibited for NMD (e.g. depleted also for UPF1).

3) The authors present evidence that the putative MIF4G domain of ICE1 is itself sufficient for mediating an interaction with EJC proteins and that over-expression of this domain can partially inhibit NMD (Figure 4). To further demonstrate of the importance and requirement of this domain and to help preclude an independent role for ICE1 in mRNA metabolism (through its activity in snRNA transcription, for example), the authors should examine the requirement of ICE1 lacking its putative MIF4G domain for interaction with EJCs and the observed reduction in NMD activity.

4) Experiments directly assessing a role for ICE1 in EJC assembly should be provided to support the main conclusion of this work. For example, the ability of UPF3B to assemble into EJCs in vitro should be evaluated in the presence and absence of ICE1 (with and without its MIF4G domain).

5) Given that the EJC composition is altered and that UPF3B is retained in the nucleus upon ICE1 depletion, the authors should provide evidence that NMD inhibition is not due to retention of mRNA in the nucleus or inhibition of translation in the cytoplasm.

6) Depletion efficiencies for the various factors are generally not reported (Figure 2D and 5A are notable exceptions) and controls are often lacking. Note that depletion of ICE1 in Figure 5A is quite poor.

*Reviewer #3:*

In a genome-wide siRNA screen using a Luciferase-based NMD reporter, Baird and colleagues identified – apart from some of the well-known NMD factors and the EJC core components – ICE1 as a new NMD factor. While the identification only one new NMD factor from such a "tour de force" approach may be somewhat disappointing, the authors did a nice job in investigating the role of ICE1 in NMD. So far, very little was known about ICE1 apart from an involvement in the assembly of the small elongation complex, which plays a role in snRNA transcription. Using a combination of knockdowns, overexpressions, RNA-seq, NMD reporter assays and IPs, the authors provide compelling evidence that ICE1 facilitates the assembly of UPF3B with the EJC core and thereby promotes EJC-enhanced NMD.

Before publication, the following two points should be addressed:

1) Figure 4A: It seems that the FLAG antibody failed to pull down the 3XF-MIF4G ICE1 construct. Instead you have a strong unspecific band (also present in the mock) that is detected with the FLAG antibody. Nevertheless, eIF4AIII is only detected in the IP of the cells expressing the ICE1 MIF4G but not in the mock. Something is not kosher with this IP; please explain.

2) Discussion: The authors state that ICE1 may be involved in degradation of a subset of 3'UTR-mediated decay targets. I wonder if this subset might be 3'UTRs that contain a spliced intron, and by inference an EJC, and thus belong to the EJC-enhanced class of NMD targets. Were the 3' UTR transcripts used for the analysis in Figure 2E filtered for transcripts lacking annotated introns in the 3' UTRs or could the observed effect originate from such "EJC-enhanced" NMD targets with 3' UTR introns? Consistent with my suggestion, the long 3' UTR of SMG5 mRNA is a NMD-inducing feature and contains no annotated intron, and this transcript was not affected by ICE1 knockdown (Figure 6B). Re-analyzing the RNA-seq data could perhaps solve this important question.

---

## [Author Response]

Reviewer #1:[…] 1) Where are the UPF proteins in Figure 1C?

The rank seed-corrected Z-scores of the siRNAs targeting ICE1 and UPF proteins among the 64,753 siRNAs in the screen are as follows:

UPF1: 218, 745, 1367

UPF2: 219, 2800, 37181

UPF3B: 1948, 5062, 9257

ICE1: 239, 342, 30797

Since it is difficult to represent all of these siRNAs in the plot in Figure 1C, we chose SMG1 and SMG6 as two well-characterized NMD factors with strong responses in the screen. The remainder of the individual siRNA data can be found in the “Primary by siRNA” tab of Supplementary file 1.

2) I found Figure 4A nearly impossible to interpret. What are the bands in the anti-flag panel blot in the anti-flag IP? Non-specific bands should be indicated as well as specific ones in both sides of the panel. It looks like the mock and 3XF-MIF4GICE1 lanes are identical, meaning that there's no evidence that the pulldown worked (or else this is a non-specific band which is lower in the 3XF-UPF3B lane). This is the only evidence for sufficiency of the MIF4G domain for interaction with eIF4AIII, so this figure needs to be clarified – and if these are the main data, the interaction is quite weak. Is MATRIN3 a control for non-specific association?

We have simplified our presentation of these experiments in the revised figure (now Figure 5A). As referenced in the previous figure legend, the high levels of UPF3B and IgG present on the immunoblot and inefficient detection due to probing of the membrane with multiple antibodies made the lower levels of recovery of the putative MIF4G domain difficult to visualize. We have resolved this issue in the revised version by focusing on the putative MIF4G domain and using exposures that more clearly show the association. We have also added Figure 5—figure supplement 1, in which we performed FLAG purifications of transiently expressed full-length ICE1 protein, a version lacking the C-terminus, and the C-terminus alone, finding that the C-terminus preferentially recovered EJC proteins.

3) In Figure 5B, the total amount of GFP-UPF3B is not presented, only the nuclear/cytoplasmic ratio. Are the overall levels of UPF3B affected?

We have added a western blot showing that levels of the GFP-UPF3B protein are not affected by ICE1 knockdown (now Figure 6B). We agree that the levels of GFP in the knockdown cells appear higher, but this is due to concentration of the protein in a smaller area of the image. We have also revised the figure legend to explain that the cell images were selected from those exhibiting the median nuclear/cytoplasmic ratio in the respective conditions.

4) For the scatterplot in Figure 3—figure supplement 1A, both axes are normalized to the same siNT control dataset, which makes them poor evidence for similarity of UPF1 and ICE1 effects. For 3 arrays generated from a normal distribution (A, B, C), log(A/C) and log(B/C) will be highly correlated most of the time.

We agree that correlation of overall changes in mRNA abundance is not the best way to illustrate the effect of ICE1 on NMD substrates. In addition to the reason cited by the reviewer, the involvement of both ICE1 and UPF1 in multiple cellular processes complicates the analysis. We have taken several approaches in the revised manuscript to further analyze the relationship between ICE1 and NMD. Using a recently developed approach to infer relative transcript stability from steady-state RNAseq, we find that mRNAs with uORFs or long 3’UTRs are systematically stabilized by ICE1 knockdown (Figure 3—figure supplement 1E and F). We also show that transcripts that exhibit increases in steady state levels upon UPF1 or ICE1 depletion are preferentially stabilized in ICE1 knockdown cells (Figure 3A and Figure 3—figure supplement 1B, C, and D). Further, we have performed a second RNAseq study involving UPF3B depletion, from which we find that mRNAs that are stabilized in both UPF1 and UPF3B depletion show larger increases in relative stability upon ICE1 depletion than those stabilized in only one knockdown condition (Figure 3B). We have also corroborated these findings by using metabolic labeling to show that ICE1 depletion increases half-lives of endogenous NMD targets (Figure 3C).

5) The last paragraph of the subsection “ICE1 depletion increases abundance of transcripts with NMD-inducing features” overinterprets the data in Figure 2E and Figure 2—figure supplement 1C. I think the simplest explanation for the differences in siICE and siUPF1 would be differences in knockdown efficiency, or perhaps differences in concentration requirements for activity. I don't think that these data represent evidence for a UTR-length dependent activity difference.

It was not our intention to argue strongly for a UTR-length dependent activity, but agree that our previous discussion could be interpreted that way. We agree with the reviewer that the origin of this difference is ambiguous and have revised our description of the effect and clarified that it could be due to biological or technical effects.

6) In Figure 4—figure supplement 1, it seems that there is some residual IP of UPF3B with ICE when UPF3B lacks the EJC-binding domain. Do these mutants display residual EJC binding? Some additional controls, perhaps with EJC factors or interacting proteins are required.

We have performed further experiments with the UPF3B mutants and do find evidence for residual interactions with the EJC (now Figure 4—figure supplement 1B).

7) The authors argue that the activity of ICE1 is entirely through recruitment of UP3B to the EJC. The ideal experiment to test this would be an epistasis experiment, which may not be possible given the essentiality of the NMD machinery. However, it might be a start to check by RNA-seq that the genome-wide effects of UPF3B knockdown on NMD are similar to the effects of ICE1 on NMD. As it is, all of the genome-wide analysis presented is with comparisons of ICE to UPF1 knockdown.

We want to first clarify that we do not claim to have shown that ICE1 *only* functions in NMD by promoting the UPF3B-EJC interaction. Instead, our data suggest that this is a role of ICE1 in NMD, but we cannot (and do not claim to) rule out the possibility of additional functions of ICE1 in NMD or indirect effects of ICE1 depletion on some targets of NMD. Further, we have revised the text to more precisely state that our model does not specify that ICE1 acts through recruitment of UPF3B to the EJC. We instead say that it promotes the UPF3B-EJC association and more clearly explain that this could either be by increased recruitment or increased stability of the assembled complex.

The reviewer is correct that loss of cell viability upon knockout of either core NMD factors or ICE1 precludes epistasis experiments. As suggested, we have instead performed RNAseq on UPF3B knockdown cells, finding that ICE1 knockdown stabilizes RNAs stabilized in UPF1 and UPF3B knockdowns to a similar extent and that the population of RNAs stabilized in both UPF1 and UPF3B knockdowns show an even greater response to ICE1 knockdown (Figure 3B). These data are consistent with the model that ICE1 affects decay of UPF1- and UPF3B-dependent targets.

8) In the Discussion: "interference with ICE1 function leaves the UPF2-UPF3B interaction intact". It is not clear to me from Figure 5A that this statement is true. The UPF3B/UPF2 interaction was not tested upon ICE1 depletion (or not presented). It seems like UPF2 may be more depleted at the EJC than UPF3b upon ICE1 depletion, but this was not quantified.

We have revised this section to clarify that this statement is based on the lack of effect of overexpression of the putative MIF4G domain on the UPF2/UPF3B interaction (Figure 6—figure supplement 1), rather than the ICE1 knockdowns now shown in Figure 6A. The lower levels of UPF2 recovered in the CASC3 IPs precluded accurate quantification, so we cannot draw conclusions about possible additional effects on UPF2-EJC association.

Reviewer #2:[…] 1) There are a number of pieces of data to suggest that ICE1 may have an independent or additional function in mRNA metabolism outside of promoting UPF3B association with EJCs.a) While depletion of ICE1 appears to cause a reproducible reduction (but not elimination) in UPF3B association with core EJC components, its depletion increases the abundance of two characterized NMD substrates 4-fold greater than depletion of the core NMD factor, UPF1 (Figure 4B). This is completely unexpected if ICE1 function on these transcripts is through NMD.

Because these experiments involve partial gene depletion, not gene knockouts (the NMD proteins and ICE1 are essential for cell viability), it is difficult to base conclusions on the relative magnitudes of the effects observed with different gene knockdowns. It is well established that some proteins require a greater extent of depletion to manifest a phenotype than others. As UPF1 is an abundant protein, others and we have observed that it is relatively insensitive to changes in dosage. On the other hand, ICE1 is less abundant and appears to work at a specific interval in mRNP maturation, making it plausible that it would be more sensitive to changes in protein levels. Illustrating this point, several NMD-related proteins produced higher Z-scores in the screen than UPF1, including SMG1, SMG5, SMG6, SMG7, and all of the core members of the EJC. Further, it is also well established that the NMD pathway is subject to extensive feedback regulation, in part through decay of NMD factor mRNAs with long 3’UTRs. In our experiments, we see a preferential impact of ICE1 depletion on EJC-stimulated decay, with little or no effect on the NMD factor mRNAs with long 3’UTRs. This may mean that the pathway is less able to compensate for the loss of ICE1 by increasing the expression of the core NMD proteins, resulting in larger changes in steady state abundance of some targets.

Also, we wish to clarify that we do not claim or believe that ICE1 does not have a role in mRNA metabolism outside of its function in promoting EJC-UPF3B interactions. The evidence in the literature is clear that it is involved in snRNA transcription as part of the little elongation complex, and we also observe alterations in splicing of some genes (Supplementary file 5 of the revised manuscript). We have extended our analysis of the RNAseq data to test whether alternative splicing of mRNAs with NMD-inducing features could explain our results. As part of this effort, we repeated our analysis of the effect of ICE1 depletion on uORFs and long 3’UTRs after removing all genes for which there was an indication of alternative splicing in response to ICE1 depletion, using two independent programs and highly permissive cutoffs, but this had no effect on our findings (see Figure 2—figure supplement 1). We also provide RNAseq traces for several validated targets (Figure 2—figure supplement 2), which indicate that the changes in abundance cannot be explained by altered processing. Together, our data suggest that ICE1 is important for promoting proper association of UPF3B with EJCs and that NMD is inhibited when this function is abrogated.

b) While over-expression of UPF3B completely restores its ability to associate with EJCs (as measured by CASC3 CoIP; Figure 6A), the abundance of several NMD-sensitive mRNAs is only partially restored (~50%; Figure 6B) – indicating that abrogation of NMD in ICE1-depleted cells is not caused entirely by impaired interaction between UPF3B and the EJC.

As discussed above, we do not claim that ICE1 does not have additional functions to those uncovered here, but instead present evidence that this is a mechanism by which it can affect NMD. Further, this comment over-interprets our findings. Because knockdowns and overexpression are by nature heterogenous from cell-to-cell, some cells may exhibit enhanced UPF3B-EJC association in response to overexpression, while others will exhibit partial or no rescue. Thus, it is not unexpected that a bulk biochemical assay would indicate complete rescue on average while a functional assay would indicate incomplete rescue. This is particularly true if the defect in UPF3B-EJC interactions is due to reduced stability of the complex. In this scenario, the association would appear to be fully rescued at steady state, but the complexes may not persist long enough to fulfill their cytoplasmic function in NMD.

c) A function for ICE1 in EJC assembly/remodeling is unexpected given its absence from past biochemical characterizations of EJC components.

We are aware of two major unbiased biochemical efforts to identify EJC components via mass spectrometry (Tange et al., *RNA*, 2005, and Singh et al., 2012). While it is true that these studies did not identify ICE1 as an EJC-interacting protein, they also failed to identify UPF3B in association with EJCs, clearly indicating that they were not exhaustive analyses of EJC-interacting proteins. Partially explaining the earlier lack of identification, we have observed that Flag-tagged eIF4A3 (as used in the Singh et al., study) is incapable of efficiently co-purifying either UPF3B or ICE1, whereas antibodies against endogenous ICE1 or eIF4A3 allow reciprocal co-immunoprecipitation.

2) It is never tested whether ICE1 function on NMD-substrate abundance is directly mediated through the NMD pathway. For example, ICE1 depletion should not alter mRNA levels in cells also inhibited for NMD (e.g. depleted also for UPF1).

As above, this type of experiment is not possible using gene knockdowns, due to the complicating factor that only partial protein depletion can be observed. Because we can only incompletely deplete these essential proteins, we would in fact predict that concurrent depletion of ICE1 and UPF1 would cause a greater defect than depletion of either protein alone. For example, Huang et al., 2011 showed that simultaneous depletion of UPF1 and SMG1 increased levels of NMD target mRNAs above those observed upon independent depletion of UPF1 or SMG1. This finding illustrates that this is not a valid criterion for evaluating the involvement of a protein in NMD. Instead, we have bolstered our analysis of the effect of ICE1 depletion on well-characterized NMD targets and other mRNAs with known NMD-inducing features. Please see the response to reviewer 1, point 4, for details. These data all indicate that ICE1 depletion affects NMD, and our biochemical studies suggest that this is at least in part due to ICE1’s ability to interact with EJCs and promote UPF3B-EJC interactions.

3) The authors present evidence that the putative MIF4G domain of ICE1 is itself sufficient for mediating an interaction with EJC proteins and that over-expression of this domain can partially inhibit NMD (Figure 4). To further demonstrate of the importance and requirement of this domain and to help preclude an independent role for ICE1 in mRNA metabolism (through its activity in snRNA transcription, for example), the authors should examine the requirement of ICE1 lacking its putative MIF4G domain for interaction with EJCs and the observed reduction in NMD activity.

We have added Figure 5—figure supplement 1, in which we performed FLAG purifications of transiently expressed full-length ICE1 protein, a version lacking the C-terminus, and the C-terminus alone, finding that the C-terminus preferentially recovered EJC proteins.

4) Experiments directly assessing a role for ICE1 in EJC assembly should be provided to support the main conclusion of this work. For example, the ability of UPF3B to assemble into EJCs in vitro should be evaluated in the presence and absence of ICE1 (with and without its MIF4G domain).

We hope to reconstitute this system in vitro in the future, but such an effort is far beyond the scope of this study.

5) Given that the EJC composition is altered and that UPF3B is retained in the nucleus upon ICE1 depletion, the authors should provide evidence that NMD inhibition is not due to retention of mRNA in the nucleus or inhibition of translation in the cytoplasm.

The effect of ICE1 depletion on the luciferase reporter indicated that mRNA export and translation is not impaired in the absence of ICE1. To further investigate this possibility, we performed immunoblotting for several ICE1-dependent NMD targets, finding that ICE1 knockdown increased protein expression along with mRNA levels (Figure 3D). A concomitant increase in protein levels illustrates the respective mRNA substrates undergo proper maturation in the nucleus and transport to the cytoplasm for mRNA translation.

6) Depletion efficiencies for the various factors are generally not reported (Figure 2D and 5A are notable exceptions) and controls are often lacking. Note that depletion of ICE1 in Figure 5A is quite poor.

We have updated the manuscript to include measures of depletion efficiency in all relevant cases, either by qPCR or immunoblotting.

Reviewer #3:[…] 1) Figure 4A: It seems that the FLAG antibody failed to pull down the 3XF-MIF4G ICE1 construct. Instead you have a strong unspecific band (also present in the mock) that is detected with the FLAG antibody. Nevertheless, eIF4AIII is only detected in the IP of the cells expressing the ICE1 MIF4G but not in the mock. Something is not kosher with this IP; please explain.

As discussed in the response to reviewer 1, we have revised our presentation of this figure (now Figure 5A) to clearly illustrate that the putative MIF4G is recovered in the IP, leading to co-purification of eIF4AIII.

2) Discussion: The authors state that ICE1 may be involved in degradation of a subset of 3'UTR-mediated decay targets. I wonder if this subset might be 3'UTRs that contain a spliced intron, and by inference an EJC, and thus belong to the EJC-enhanced class of NMD targets. Were the 3' UTR transcripts used for the analysis in Figure 2E filtered for transcripts lacking annotated introns in the 3' UTRs or could the observed effect originate from such "EJC-enhanced" NMD targets with 3' UTR introns? Consistent with my suggestion, the long 3' UTR of SMG5 mRNA is a NMD-inducing feature and contains no annotated intron, and this transcript was not affected by ICE1 knockdown (Figure 6B). Re-analyzing the RNA-seq data could perhaps solve this important question.

We tested this possibility by removing all genes for which the RNAseq showed evidence for isoforms containing introns downstream of termination codons (the set of genes examined in Figure 2A) from the analysis, but this had no effect on the enhanced expression of long 3’UTR-containing mRNAs (Figure 2—figure supplement 1). Further, manual examination of RNAseq reads arising from the long 3’UTR-containing mRNAs studied in Figure 2F (and others) revealed no evidence for downstream splicing events. Instead, we hypothesize that the effect of ICE1 depletion on long 3’UTRs is due to reduced levels of UPF3B in the cytoplasm.